# Fast Optimal Locally Private Mean Estimation via Random Projections

**Hilal Asi**
Apple Inc.
hilal.asi94@gmail.com

**Vitaly Feldman**
Apple Inc.
vitaly.edu@gmail.com

**Jelani Nelson**
UC Berkeley
minilek@berkeley.edu

**Huy L. Nguyen**
Northeastern University
hu.nguyen@northeastern.edu

**Kunal Talwar**
Apple Inc.
kunal@kunaltalwar.org

## Abstract

We study the problem of locally private mean estimation of high-dimensional vectors in the Euclidean ball. Existing algorithms for this problem either incur suboptimal error or have high communication and/or run-time complexity. We propose a new algorithmic framework, ProjUnit, for private mean estimation that yields algorithms that are computationally efficient, have low communication complexity, and incur optimal error up to a $1 + o(1)$-factor. Our framework is deceptively simple: each randomizer projects its input to a random low-dimensional subspace, normalizes the result, and then runs an optimal algorithm such as PrivUnitG in the lower-dimensional space. In addition, we show that, by appropriately correlating the random projection matrices across devices, we can achieve fast server run-time. We mathematically analyze the error of the algorithm in terms of properties of the random projections, and study two instantiations. Lastly, our experiments for private mean estimation and private federated learning demonstrate that our algorithms empirically obtain nearly the same utility as optimal ones while having significantly lower communication and computational cost.

## 1 Introduction

Distributed estimation of the mean, or equivalently the sum, of vectors $v_1, \ldots, v_n \in \mathbb{R}^d$ is a fundamental problem in distributed optimization and federated learning. For example, in the latter, each of $n$ devices may compute some update to parameters of a machine learning model based on its local data, at which point a central server wishes to apply all updates to the model, i.e. add $\sum_{i=1}^n v_i$ to the vector of parameters. The typical desire to keep local data private necessitates methods for computing this sum while preserving privacy of the local data on each individual device, so that the central server essentially only learns the noisy sum and (almost) nothing about each individual summand $v_i$ [20, 11].

The gold standard for measuring privacy preservation is via the language of *differential privacy* [21]. In this work, we study this problem in the setting of *local differential privacy* (LDP). We consider (one-round) protocols for which there exists some randomized algorithm $\mathcal{R} : \mathbb{R}^d \to \mathcal{M}$ (called the *local randomizer*), such that each device $i$ sends $\mathcal{R}(v_i)$ to the server. We say the protocol is $\varepsilon$-*differentially private* if for any $v, v' \in \mathbb{R}^d$ and any event $S \subseteq \mathcal{M}$, $\Pr(\mathcal{R}(v) \in S) \leq e^\varepsilon \Pr(\mathcal{R}(v') \in S)$. If $\varepsilon = 0$ then the distribution of $\mathcal{R}(v)$ is independent of $v$, and hence the output of $\mathcal{R}(\cdot)$ reveals nothing about the local data (perfect privacy); meanwhile if $\varepsilon = \infty$ then the distributions of $\mathcal{R}(v)$ and $\mathcal{R}(v')$ can be arbitrarily far, so that in fact one may simply set $\mathcal{R}(x) = x$ and reveal local data in the clear (total lack of privacy). Thus, $\varepsilon \geq 0$ is typically called the *privacy loss* of a protocol.

37th Conference on Neural Information Processing Systems (NeurIPS 2023).

There has been much previous work on private protocols for estimating the mean $\mu := \frac{1}{n}\sum_{i=1}^{n} v_i$ in the LDP setting. Henceforth we assume each $v_i$ lives on the unit Euclidean sphere [1] $\mathbb{S}_{d-1} \subset \mathbb{R}^d$. Let $\hat{\mu}$ be the estimate computed by the central server based on the randomized messages $\mathcal{R}(v_1), \ldots, \mathcal{R}(v_n)$ it receives. Duchi and Rogers [18] showed that the asymptotically optimal expected mean squared error $\mathbb{E}\|\mu - \hat{\mu}\|_2^2$ achievable by any one-round protocol must be at least $\Omega(\frac{d}{n\min(\varepsilon,\varepsilon^2)})$, which is in fact achieved by several protocols [19, 9, 14, 24]. These protocols however achieved empirically different errors, with some having noticeably better constant factors than others.

Recent work of Asi et al. [7] sought to understand the optimal error achievable by any protocol. Let $\mathcal{R}$ be any local randomizer satisfying $\varepsilon$-DP, and $\mathcal{A}$ be an aggregation algorithm for the central server such that it computes its mean estimate as $\hat{\mu} := \mathcal{A}(\mathcal{R}(v_1), \ldots, \mathcal{R}(v_n))$. Furthermore, suppose that the protocol is *unbiased*, so that $\mathbb{E}\hat{\mu} = \mu$ for any inputs $v_1, \ldots, v_n$. Lastly, let $\mathcal{A}_{\mathrm{PrivUnit}_\varepsilon}, \mathcal{R}_{\mathrm{PrivUnit}_\varepsilon}$ denote the PrivUnit$_\varepsilon$ protocol of [9] (parameterized to satisfy $\varepsilon$-DP [2]). Let $\mathrm{Err}_{n,d}(\mathcal{A}, \mathcal{R})$ denote

$$\sup_{v_1,\ldots,v_n \in S^{d-1}} \|\mathcal{A}(\mathcal{R}(v_1), \ldots, \mathcal{R}(v_n)) - \mu\|_2^2.$$

Asi et al. [7] proved the remarkable theorem that for any $n, d$:

$$\mathrm{Err}_{n,d}(\mathcal{A}, \mathcal{R}) \geq \mathrm{Err}_{n,d}(\mathcal{A}_{\mathrm{PrivUnit}_\varepsilon}, \mathcal{R}_{\mathrm{PrivUnit}_\varepsilon}).$$

Thus, PrivUnit is not only asymptotically optimal, but in fact actually optimal in a very strong sense (at least, amongst unbiased protocols).

While this work thus characterizes the optimal error achievable for $\varepsilon$-LDP mean estimation, there are other desiderata that are important in practice. The most important amongst them are the *device runtime* (the time to compute $\mathcal{R}(v)$), the *server runtime* (the time to compute $\mathcal{A}$ on $(\mathcal{R}(v_1), \ldots, \mathcal{R}(v_n))$), and the *communication cost* ($\lceil \log_2 |\mathcal{M}| \rceil$ bits for each device to send its $\mathcal{R}(v)$ to the server). The known error-optimal algorithms (PrivUnit [9] and PrivUnitG [7]) either require communicating $d$ floats or have a slower device runtime of $\Omega(e^\varepsilon d)$. As mean estimation is often used as a subroutine in high-dimensional learning settings, this communication cost can be prohibitive and this has led to a large body of work on reducing communication cost [2, 27, 14, 26, 24, 12]. Server runtimes of these optimal algorithms are also slow, scaling as $nd$, whereas one could hope for nearly linear time $\tilde{O}(n + d)$ (see Table 1).

Chen et al. [14] recently studied this tradeoff and proposed an algorithm called SQKR, which has an optimal communication cost of $\varepsilon$ bits and device runtime only $O(d \log^2 d)$. However, this algorithm incurs error that is suboptimal by a constant factor, which can be detrimental in practice. Indeed our experiments in Section 4 demonstrate the significance of such constants as the utility of these algorithms does not match that of optimal algorithms even empirically, resulting for example in 10% degradation in accuracy for private federated learning over MNIST with $\varepsilon = 10$.

Feldman and Talwar [24] give a general approach to reducing communication via rejection sampling. When applied to PrivUnitG, it yields a natural algorithm that we call Compressed PrivUnitG. While it yields optimal error and near-optimal communication, it requires device run time that is $O(e^\varepsilon d)$. These algorithms are often used for large $d$ (e.g. in the range $10^5 - 10^7$) corresponding to large model sizes. The values of $\varepsilon$ are often in the range 4-12 or more, which may be justifiable due to privacy being improved by aggregation or shuffling [10, 15, 22, 25]. For this range of values, the $\Theta(e^\varepsilon d)$ device runtime is prohibitively large and natural approaches to reduce this in Feldman and Talwar [24] lead to increased error. To summarize, in the high-dimensional setting, communication-efficient local randomizers are forced to choose between high device runtime or suboptimal error (see Table 1).

Another related line of work is non-private communication efficient distributed mean estimation where numerous papers have recently studied the problem due to its importance in federated learning [37, 4, 29, 2, 26, 23, 32, 38, 39]. Similarly to our paper, multiple works in this line of work have used random rotations to design efficient algorithms [37–39]. However, the purpose of these works is to

---

[1]One often considers the problem for the vectors being of norm at most 1, rather than exactly 1. It is easy to show that vectors $v$ in the unit ball in $\mathbb{R}^d$ can be mapped to $\mathbb{S}_d \subseteq \mathbb{R}^{d+1}$, simply as $(v, 1 - \|v\|_2^2)$. Thus up to changing $d$ to $d + 1$, the two problems are the same. Since we are interested in the case of large $d$, we choose the version that is easier to work with.

[2]There are multiple ways to set parameters of PrivUnit to achieve $\varepsilon$-DP; we assume the setting described by Asi et al. [7], which optimizes the parameters to minimize the expected mean squared error.

develop better quantization schemes for real-valued vectors to reduce communication to $(1 + o(1)) \cdot d$ bits. This is different from our goal, which is to send $k \ll d$ parameters while still obtaining the statistically optimal bounds up to a $1 + o(1)$ factor. Moreover, in order to preserve privacy, our algorithms require new techniques to handle the norms of the projected vectors, and post-process them using different normalization schemes, in order to guarantee that the projection error after post-processing is negligible.

## 1.1 Contributions

Our main contribution is a new framework, ProjUnit, for private mean estimation which results in near-optimal, efficient, and low-communication algorithms. Our algorithm obtains the same optimal utility as PrivUnit and PrivUnitG but with a significantly lower polylogarithmic communication complexity, and device runtime that is $O(d \log d)$ and server runtime $\tilde{O}(n + d)$. We also implement our algorithms and show that both the computational cost and communication cost are small empirically as well. Figure 1 plots the error as a function of $\varepsilon$ for several algorithms and demonstrates the superiority of our algorithms compared to existing low-communication algorithms (see more details in Section 4). Moreover, we show that the optimal error bound indeed translates to fast convergence in our private federated learning simulation.

At a high level, each local randomizer in our algorithm first projects the vector to a randomly chosen lower-dimensional subspace, and then runs an optimal local randomizer in this lower-dimensional space. At first glance, this is reminiscent of the use of random projections in the Johnson-Lindenstrauss (JL) transform or the use of various embeddings in prior work (such as [14]). However, unlike the JL transform and embeddings in prior work, in our application, each point uses its own projection matrix . The JL transform is designed to preserve *distances* between points, not the points themselves. In our application, a random projection is used to obtain a low-dimensional unbiased estimator for a point; however the variance of this estimator is quite large (of the order of $d/k$). Our main observation is that while large, this variance is small compared to the variance added due to privacy when $k$ is chosen appropriately. This fact allows us to use the projections as a pre-processing step. With some care needed to control the norm of the projected vector which is no longer fixed, we then run a local randomizer in the lower dimensional space. We analyze the expected squared error of the whole process and show that as long as the random projection ensemble satisfies certain specific properties, the expected squared error of our algorithm is within $(1 + o(1))$ factors of the optimal; the $o(1)$ term here falls with the projection dimension $k$. The required properties are easily shown to be satisfied by random orthogonal projections. We further show that more structured projection ensembles, that allow for faster projection algorithms, also satisfy the required properties, and this yields even faster device runtime.

Although these structured projections result in fast device runtime, they still incur an expensive computational cost for the server which needs to apply the inverse transformation for each client, resulting in runtime $O(nd \log d)$. Specifically, each device sends a privated version $\hat{v}_i$ of $W_i v_i$, and the server must then compute $\sum_i W^\top \hat{v}_i$. To address this, we use correlated transformations in order to reduce server runtime while maintaining optimal accuracy up to $1 + o(1)$ factors. In our correlated ProjUnit protocol, the server pre-defines a random transformation $W$, which all devices then use to define $W_i = S_i W$ where $S_i$ is a sampling matrix. The advantage is then that $\sum_i W_i^\top \hat{v}_i$ is replaced with $\sum_i W^\top \hat{v}_i = W^\top (\sum_i S_i \hat{v}_i)$, which can be computed more quickly as it requires only a single matrix-vector multiplication. The main challenge with correlated transformations is that the correlated client transformations result in increased variance. However, we show that the independence in choosing the sampling matrices $S_i$ is sufficient to obtain optimal error.

Finally, we note without correlating projections each client using its own projection would imply that each projection needs to be communicated to the server. Doing this naively would require communicating $kd$ real values completely defeating the benefits of our protocol. However the projection matrix does not depend on the input and therefore can be communicated cheaply using a seed to an appropriate pseudorandom generator.

## 2 A random projection framework for low-communication private algorithms

In this section we propose a new algorithm, namely ProjUnit, which has low communication complexity and obtains near-optimal error (namely, up to a $1 + o(1)$ factor of optimum). The starting point

| | Utility | Run-time (client) | Run-time (server) | Communication |
|---|---|---|---|---|
| Repeated PrivHS [19, 24] | $O(\mathsf{OPT})$ | $\varepsilon d$ | $n \lceil \varepsilon \rceil$ | $\varepsilon \cdot \mathsf{poly}(\log d)$ |
| PrivUnit [9] | $\mathsf{OPT}$ | $d$ | $nd$ | $d$ |
| SQKR [14] | $O(\mathsf{OPT})$ | $d \log^2 d$ | $n\varepsilon \log d + d \log^2 d$ | $\varepsilon \log d$ |
| CompPrivUnit [24] | $(1 + o(1)) \cdot \mathsf{OPT}$ | $e^\varepsilon d$ | $nd \log d$ | $\varepsilon \cdot \mathsf{poly}(\log d)$ |
| FastProjUnit (Section 2) | $(1 + o(1)) \cdot \mathsf{OPT}$ | $d \log d$ | $nd \log d$ | $\varepsilon \log^2 d$ |
| FastProjUnit-corr (Section 3) | $(1 + o(1)) \cdot \mathsf{OPT}$ | $d \log d$ | $n \log^3 d + n\varepsilon \log d + d \log d$ | $\varepsilon \log^2 d$ |

Table 1: Comparison of Error-Runtime-Communication trade-offs for different algorithms for private mean estimation. The last two rows use our algorithms from Section 2 and Section 3 with a communication budget $k \approx \varepsilon \log d$. PrivUnitG is omitted as it has the same guarantees as PrivUnit except utility where it has $(1 + o(1)) \cdot \mathsf{OPT}$. We omit constant factors from the run-time and communication complexities.

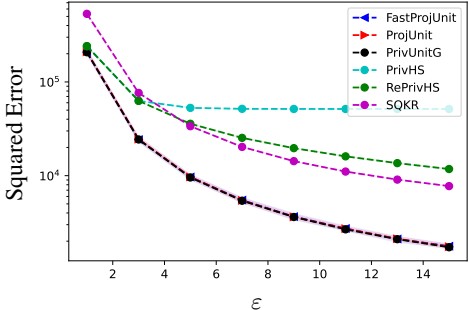

Figure 1: Squared error of different algorithms as a function of $\varepsilon$ for $d = 32768$ averaged over 50 runs with 90% confidence intervals. The lines for the top three algorithms are almost identical.

of our algorithms is a randomized projection map in $\mathbb{R}^{k \times d}$ which we use to project the input vectors to a lower-dimensional space. The algorithm then normalizes the vector as a necessary pre-processing step. Finally, the local randomizer applies PrivUnitG [7] (see Algorithm 8 in Appendix) over the normalized projected vector and sends the response to the server. The server then applies the inverse transformation and aggregates the responses in order to estimate the mean. We present the full details of the client and server algorithms in Algorithm 1 and Algorithm 2.

To analyze this algorithm, we first present our general framework for an arbitrary distribution over projections. In the next sections we utilize different instances of the framework using different random projections such as random rotations and more structured transforms.

---

**Algorithm 1** ProjUnit (client)

---

**Require:** Input vector $v \in \mathbb{R}^d$, Distribution over projections $\mathcal{W}$.
 1: Randomly sample transform $W \in \mathbb{R}^{k \times d}$ from $\mathcal{W}$
 2: Project the input vector $v_p = Wv$
 3: Normalize $u = \frac{v_p}{\|v_p\|_2}$
 4: Let $\hat{u} = \mathrm{PrivUnitG}(u)$ (as in Algorithm 8)
 5: Send $\hat{u}$ and (encoding of) $W$ to server

---

The following theorem states the privacy and utility guarantees of ProjUnit for a general distribution over transformation $\mathcal{W}$ that satisfies certain properties. For ease of notation, let $\mathcal{R}_{\mathsf{PU}}$ denote the ProjUnit local randomizer of the client (Algorithm 1), and $\mathcal{A}_{\mathsf{PU}}$ denote the server aggregation of

---

**Algorithm 2** ProjUnit (server)

---
1: Receive $\hat{u}_1, \ldots, \hat{u}_1$ from clients with (encodings of) transforms $W_1, \ldots, W_n$
2: Return the estimate $\hat{\mu} = \frac{1}{n} \sum_{i=1}^n W_i^\top \hat{u}_i$

---

ProjUnit (Algorithm 2). To simplify notation, we let

$$\mathsf{Err}_{n,d}(\text{PrivUnitG}) = \mathsf{Err}_{n,d}(\mathcal{A}_{\text{PrivUnitG}_\varepsilon}, \mathcal{R}_{\text{PrivUnitG}_\varepsilon}) = c_{d,\varepsilon} \frac{d}{n\varepsilon}$$

denote the error of the PrivUnitG $\varepsilon$-DP protocol where $A_{\text{PrivUnitG}_\varepsilon}, \mathcal{R}_{\text{PrivUnitG}_\varepsilon}$ denote the PrivUnitG protocol with optimized parameters (see Algorithm 8) and $c_{d,\varepsilon} = O(1)$ is a constant [7]. We defer the proof to Appendix B.1.

**Theorem 1.** *Let $k \leq d$ and assume that the transformations $W_i \in \mathbb{R}^{k \times d}$ are independently chosen from a distribution $\mathcal{W}$ that satisfies:*

1. *Bounded operator norm: $\mathbb{E}\left[\left\|W_i^\top\right\|^2\right] \leq d/k + \beta_\mathcal{W}$.*

2. *Bounded bias: $\left\|\mathbb{E}\left[\frac{W_i^\top W_i v}{\|W_i v\|_2}\right] - v\right\|_2 \leq \sqrt{\alpha_\mathcal{W}}$ for all unit vectors $v \in \mathbb{R}^d$.*

*Then for all unit vectors $v_1, \ldots, v_n \in \mathbb{R}^d$, setting $\hat{\mu} = \mathcal{A}_{\mathsf{PU}}\left(\mathcal{R}_{\mathsf{PU}}(v_1), \ldots, \mathcal{R}_{\mathsf{PU}}(v_n)\right)$, the local randomizers $\mathcal{R}_{\mathsf{PU}}$ are $\varepsilon$-DP and*

$$\mathbb{E}\left[\left\|\hat{\mu} - \frac{1}{n} \sum_{i=1}^n v_i\right\|_2^2\right] \leq \mathsf{Err}_{n,d}(\textit{PrivUnitG}) \cdot \left(1 + \frac{k\beta_\mathcal{W}}{d} + O\left(\frac{\varepsilon + \log k}{k}\right)\right) + \alpha_\mathcal{W}.$$

### 2.1 ProjUnit using Random Rotations

Building on the randomized projection framework of the previous section, in this section we instantiate it with a random rotation matrix. In particular, we sample $W \in \mathbb{R}^{k \times d}$ with the structure

$$W_H = \sqrt{\frac{d}{k}} SU, \tag{1}$$

where $U \in \mathbb{R}^{d \times d}$ is a random rotation matrix such that $U^\top U = I$, and $S \in \mathbb{R}^{k \times d}$ is a sampling matrix where each row has a single 1 in a uniformly random location (without repetitions). The following theorem states our guarantees for this distribution.

**Theorem 2.** *Let $k \leq d$ and $W \in \mathbb{R}^{k \times d}$ be a random rotation matrix sampled as described in* (1). *Then for all unit vectors $v_1, \ldots, v_n \in \mathbb{R}^d$, setting $\hat{\mu} = \mathcal{A}_{\mathsf{PU}}\left(\mathcal{R}_{\mathsf{PU}}(v_1), \ldots, \mathcal{R}_{\mathsf{PU}}(v_n)\right)$, the local randomizers $\mathcal{R}_{\mathsf{PU}}$ are $\varepsilon$-DP and*

$$\mathbb{E}\left[\left\|\hat{\mu} - \frac{1}{n} \sum_{i=1}^n v_i\right\|_2^2\right] \leq \mathsf{Err}_{n,d}(\textit{PrivUnitG}) \cdot \left(1 + O\left(\frac{\varepsilon + \log k}{k}\right)\right) + O\left(\frac{1}{k^2}\right).$$

The proof follows directly from Theorem 1 and the following proposition which proves certain properties of random rotations. We defer the proof to Appendix B.2.

**Proposition 1.** *Let $W \in \mathbb{R}^{k \times d}$ be a random rotation matrix sampled as described in* (1). *Then*

1. *Bounded operator norm: $\|W^\top\| \leq \sqrt{\frac{d}{k}}$.*

2. *Bounded bias: for every unit vector $v \in \mathbb{R}^d$: $\left\|\mathbb{E}\frac{W^\top W v}{\|W v\|} - v\right\| = O(1/k)$.*

We also have similar analysis for Gaussian transforms with an additional $O(\sqrt{k/d})$ factor in the first term. We include the analysis in Appendix D.

## 2.2 Fast ProjUnit using the SRHT

While the random rotation based randomizer enjoys near-optimal error and low communication complexity, its runtime complexity is somewhat unsatisfactory as it requires calculating $Wv$ for $W \in \mathbb{R}^{k \times d}$, taking time $O(kd)$. In this section, we propose a ProjUnit algorithm using the Subsampled Randomized Hadamard transform (SRHT), which is closely related to the fast JL transform [3]. We show that this algorithm has the same optimality and low-communication guarantees as the random rotations version, and additionally has an efficient implementation with $O(d \log d)$ client runtime.

The SRHT ensemble contains matrices $W_H \in \mathbb{R}^{k \times d}$ with the following structure:

$$W_H = \sqrt{\frac{d}{k}} SHD, \tag{2}$$

where $S \in \mathbb{R}^{k \times d}$ is a sampling matrix where each row has a single $1$ in a uniformly random location sampled without replacement, $H \in \mathbb{R}^{d \times d}$ is the Hadamard matrix, and $D \in \mathbb{R}^{d \times d}$ is a diagonal matrix where $D_{ii}$ are independent samples from the Rademacher distribution, that is, $D_{ii} \sim \mathsf{Unif}\{-1, +1\}$. The main advantage of the SRHT is that there exist efficient algorithms for matrix-vector multiplication with $H$.

The following theorem presents our main guarantees for the SRHT-based ProjUnit algorithm.

**Theorem 3.** *Let $k \leq d$ and $W$ be sampled from the SRHT ensemble as described in* (2)*. Then for all unit vectors $v_1, \ldots, v_n \in \mathbb{R}^d$, setting $\hat{\mu} = \mathcal{A}_{\mathsf{PU}}(\mathcal{R}_{\mathsf{PU}}(v_1), \ldots, \mathcal{R}_{\mathsf{PU}}(v_n))$, the local randomizers $\mathcal{R}_{\mathsf{PU}}$ are $\varepsilon$-DP and*

$$\mathbb{E}\left[\left\|\hat{\mu} - \frac{1}{n}\sum_{i=1}^n v_i\right\|_2^2\right] \leq \mathsf{Err}_{n,d}(PrivUnitG) \cdot \left(1 + O\left(\frac{\varepsilon + \log k}{k}\right)\right) + O\left(\frac{\log^2 d}{k}\right).$$

**Remark 1.** *The communication complexity of SRHT-based ProjUnit can be reduced to $O(k \log d + \log^2 d)$. To see this, note that $\hat{u}$ is a $k$-dimensional vector. Moreover, the matrix $W = \sqrt{d/k} \cdot SHD$ can be sent in $O(k \log d)$ as follows: $S$ has $k$ rows, each with a single entry that contains $1$, hence we can send the indices for each row using $\log d$ bits for each row. Moreover, $H$ is the Hadamard transform and need not be sent. Finally, $D$ is a diagonal matrix with entries $D_{ii} \sim \mathsf{Unif}\{-1, +1\}$. By standard techniques [36, 5], we only need the entries of $D$ to be $O(\log(d))$-wise independent for Proposition 2 to hold. Thus $O(\log^2 d)$ bits suffice to communicate a sampled $D$.*

The proof of the theorem builds directly on the following two properties of the SHRT.

**Proposition 2.** *Let $W$ be sampled from the SRHT ensemble as described in* (2)*. Then we have*

1. *Bounded operator norm:* $\mathbb{E}\left[\left\|W^\top\right\|\right] = \mathbb{E}\left[\|W\|\right] \leq \sqrt{d/k}$.

2. *Bounded bias: for every unit vector $v \in \mathbb{R}^d$:* $\left\|\mathbb{E}\frac{W^\top W v}{\|Wv\|} - v\right\| = O(\log(d)/\sqrt{k})$.

Theorem 3 now follows directly from the bounds of Theorem 1 using $\alpha_{\mathcal{W}} = O(\log^2(d)/k)$. We defer the proof to Appendix B.3.

**Remark 2.** *While our randomizers in this section pay an additive term that does not decrease with $n$ (e.g. $\log^2(d)/k$ for SRHT), this term is negligible in most settings of interest. Indeed, using Theorem 3 and the fact that $\mathsf{Err}_{n,d}(PrivUnitG) = c_{d,\varepsilon}d/n\varepsilon$, we get that the final error of our SRHT algorithm is roughly $c_{d,\varepsilon}d/n\varepsilon(1 + o(1)) + O(\log^2(d)/k)$. This implies that in the high-dimensional setting the bias term is negligible.*

*However, to cover the whole spectrum of parameters, we develop a nearly unbiased versions of these algorithms in Appendix A. In particular, we show in Theorem 6 that our unbiased version has error*

$$\mathsf{Err}_{n,d}(PrivUnitG) \cdot \left(1 + O\left(\frac{\varepsilon + \log k}{k} + \sqrt{\frac{\log(nd/k)}{k}}\right)\right).$$

## 3 Efficient Server Runtime via Correlated Sampling

One downside of the algorithms in the previous section is that server runtime can be costly: indeed, as each client uses an independent transformation, the server has to apply the inverse transformation

(matrix multiplication) for each client, resulting in runtime $O(nd \log d)$. In this section, we propose a new protocol that significantly reduces server runtime to $O(n \log^3 d + d \log d + nk)$ while retaining similar error guarantees. The protocol uses correlated transformations between users which allows the server to apply an inverse transformation only a small number of times. However, clients cannot use the same transformation as this will result in large bias.

The protocol works as follows: the server samples $U \in \mathbb{R}^{d \times d}$ from the Randomized Hadamard transform: $U = HD$ where $H \in \mathbb{R}^{d \times d}$ is the Hadamard transform, and $D \in \mathbb{R}^{d \times d}$ is a diagonal matrix where each diagonal entry is independently sampled from the Rademacher distribution. Then, client $i \in [n]$, samples a random sampling matrix $S_i \in \mathbb{R}^{k \times d}$, and uses $U$ to define the transform $W_i \in \mathbb{R}^{k \times d}$:

$$W_i = \sqrt{\frac{d}{k}} S_i U. \tag{3}$$

We describe the full details of the client and server algorithms for correlated ProjUnit in Algorithm 3 and Algorithm 4, and denote them $\mathcal{R}_{\mathsf{CPU}}$ and $\mathcal{A}_{\mathsf{CPU}}$, respectively. We have the following guarantee. We defer the proof to Appendix C.

---

**Algorithm 3** Correlated ProjUnit (client)

---

**Require:** Input vector $v \in \mathbb{R}^d$.
 1: Randomly sample diagonal $D$ from the Rademacher distribution based on predefined seed
 2: Sample $S \in \mathbb{R}^{k \times d}$ where each row is chosen uniformly at random without replacement from standard basis vectors $\{e_1, \ldots, e_d\}$
 3: Project the input vector $v_p = SHDv$ where $H \in \mathbb{R}^{d \times d}$ is the Hadamard matrix
 4: Normalize $u = \frac{v_p}{\|v_p\|_2}$
 5: Let $\hat{u} = \text{PrivUnitG}(u)$ (as in Algorithm 8)
 6: Send $\hat{u}$, and (an encoding of) $S$ to server

---

**Algorithm 4** Correlated ProjUnit (server)

---

 1: Receive $\hat{u}_1, \ldots, \hat{u}_1$ from clients with (encodings of) transforms $S_1, \ldots, S_n$
 2: Sample diagonal matrices $D$ from Rademacher distribution based on predefined seed
 3: Let $U = HD$ where $H \in \mathbb{R}^{d \times d}$ is the Hadamard matrix
 4: Return the estimate $\hat{\mu} = \frac{1}{n} U^\top \sum_{i=1}^n S_i^\top \hat{u}_i$

---

**Theorem 4.** *Let* $k \leq d$. *Then for all unit vectors* $v_1, \ldots, v_n \in \mathbb{R}^d$, *setting* $\hat{\mu} = \mathcal{A}_{\mathsf{CPU}}(\mathcal{R}_{\mathsf{CPU}}(v_1), \ldots, \mathcal{R}_{\mathsf{CPU}}(v_n))$, *the local randomizers* $\mathcal{R}_{\mathsf{CPU}}$ *are* $\varepsilon$-DP *and*

$$\mathbb{E}\left[\left\|\hat{\mu} - \frac{1}{n}\sum_{i=1}^n v_i\right\|_2^2\right] \leq \mathsf{Err}_{n,d}(\textit{PrivUnitG})\left(1 + O\left(\frac{\varepsilon + \log k}{k}\right)\right) + O\left(\frac{\log^2 d}{k}\right).$$

*Moreover, server runtime is* $O(n \log(d) \log^2(nd) + d \log d + nk)$.

## 4 Experiments

We conclude the paper with several experiments that demonstrate the performance of our proposed algorithms, comparing them to existing algorithms in the literature. We conduct our experiments in two different settings: the first is synthetic data, where we aim to test our algorithms and understand their performance for our basic task of private mean estimation, comparing them to other algorithms. In our second setting, we seek to evaluate the performance of our algorithms for *private federated learning* which requires private mean estimation as a subroutine for DP-SGD.

### 4.1 Private mean estimation

In our synthetic-data experiment, we study the basic private mean estimation problem, aspiring to investigate the following aspects:

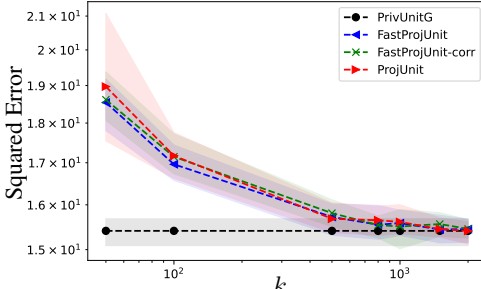

Figure 2: Performance of ProjUnit, FastProjUnit and their correlated versions with 90% confidence intervals as a function of $k$ for $d = 32768$, NumRep $= 30$, $n = 50$, and $\varepsilon = 10$

1. Utility of ProjUnit algorithms as a function of the communication budget

2. Utility of our low-communication algorithms compared to the optimal utility and other existing low-communication algorithms

3. Run-time complexity of our algorithms compared to existing algorithms

Our experiments[3] measure the error of different algorithms for estimating the mean of a dataset. To this end, we sample unit vectors $v_1, \ldots, v_n \in \mathcal{R}^d$ by normalizing samples from the normal distribution $\mathsf{N}(\mu, 1/d)$ (where $\mu \in \mathbb{R}^d$ is a random unit vector), and apply a certain privacy protocol NumRep times to estimate the mean $\sum_{i=1}^{n} v_i/n$, producing mean squared errors $e_1, \ldots, e_{\mathsf{NumRep}}$. Our final error estimate is then the mean $\frac{1}{\mathsf{NumRep}} \sum_{i=1}^{\mathsf{NumRep}} e_i$. We test the performance of several algorithms: ProjUnit (Subsection 2.1), FastProjUnit (Subsection 2.2), FastProjUnit-corr (Section 3), PrivUnitG [7], CompPrivUnitG [7, 24], PrivHS [19], RePrivHS [19, 24], SQKR [14].

In Figure 2, we plot the error for our ProjUnit algorithms as a function of the communication budget $k$. We consider a high-dimensional regime where $d = 2^{15}$ with a small number of users $n = 50$ and a bounded communication budget $k \in [1, 2000]$. Our plot shows that our ProjUnit algorithms obtain the same near-optimal error as PrivUnitG for $k$ as small as $1000$. Moreover, the plots show that the correlated versions of our ProjUnit algorithms obtain nearly the same error.

To translate this into concrete numbers, the transform $W$ can be communicated using a small seed ($\sim 128$ bits) in practice, or using $k \log d + \log^2 d \sim 20400$ bits or less than 3kB for $d = 10^6$. Sending the $k$-dimensional vector of 32-bit floats would need an additional 4kB. Thus the total communication cost is between 4 and 8kB. This can be further reduced by using a compressed version of PrivUnitG in the projected space, which requires the client to send a 128-bit seed. In this setting, the communication cost is a total of 256 bits. Thus in the sequel, we primarily focus on the $k = 1000$ version of our algorithms.

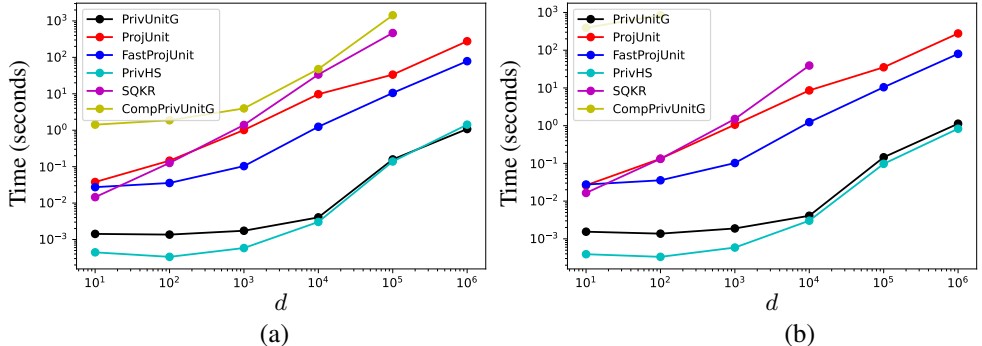

Figure 3: Run-time (in seconds) as a function of the dimension for (a) $\varepsilon = 10$ and (b) $\varepsilon = 16$. The plots for some algorithms are not complete as they did not finish within the cut-off time.

---

[3]The code is also available online on `https://github.com/apple/ml-projunit`

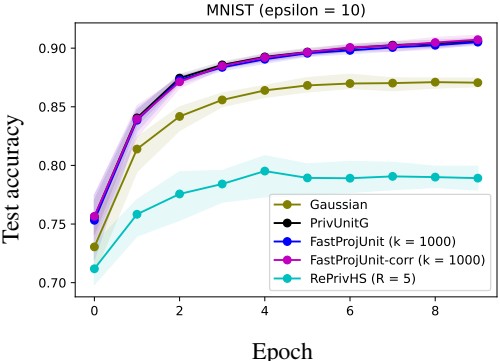

Figure 4: Test accuracy on the MNIST dataset with 90% confidence intervals as a function of epoch for $\varepsilon = 10.0$.

Next, we compare the performance of our low-communication algorithms against existing low-communication algorithms: PrivHS and SQKR. In Figure 1, we plot the error as a function of the privacy parameter for each algorithm using the best choice of $k$ (bound on communication) for each algorithm. In particular, we choose $k = \varepsilon$ for SQKR, num repetitions $R = \varepsilon/2$ for repeated PrivHS, $k = 1000$ for ProjUnit and FastProjUnit. Moreover, in this experiment we set $n = 1$ and NumRep $= 50$ to estimate the variance of each method. The figure shows that PrivHS and SQKR, while having low-communication complexity, suffer a significantly worse utility than (near-optimal) PrivUnitG. On the other side, both our ProjUnit algorithms obtain nearly the same error as PrivUnitG with a bounded communication budget of $k = 1000$.

In our third experiment in Figure 3, we plot the runtime of each algorithm as a function of the privacy parameter. Here, we use $n = 1$, NumRep $= 10$ and measure the run-time of each method for different values of the dimension $d$ and privacy parameter $\varepsilon$, allowing each method to run for 1 hour before interrupting. As expected from our theoretical analysis, the runtime of ProjUnit using random rotations is noticeably slower than the (high communication cost) PrivUnitG. However, our SRHT-based FastProjUnit is substantially faster and has a comparable run-time to PrivUnitG. Moreover, for large $\varepsilon$ and $d$, the run-time of compressed PrivUnitG becomes too costly compared to our algorithms due to the $e^{\varepsilon} d$ time complexity.

## 4.2 Private federated learning

Having demonstrated the effectiveness of our methods for private mean estimation, in this section we illustrate the improvements offered by our algorithms for private federated learning. Similarly to the experimental setup in [13], we consider the MNIST [31] dataset and train a convolutional network (see Table 2) using DP-SGD [1] with 10 epochs and different sub-routines for privately estimating the mean of gradients at each batch. In order to bound the sensitivity, we clip the gradients to have $\ell_2$-norm 1, and run DP-SGD with batch size of 600, step-size equal to 0.1, and momentum of 0.5.

Figure 4 shows our results for this experiment, where we plot the test accuracy as a function of the epoch for each method. The plots demonstrate that our ProjUnit algorithms obtain similar performance to PrivUnitG, and better performance than the Gaussian mechanism or PrivHS. For the Gaussian mechanism, we set $\delta = 10^{-5}$ and add noise to satisfy $(\varepsilon, \delta)$-DP using the analysis in [8]. We did not run SQKR in this experiment as it is not sufficiently computationally efficient for this experiment and has substantially worse performance in the experiments of the previous section. We also did not run the MVU mechanism [13] as their experiments show that it is worse than the Gaussian mechanism which has worse performance than our methods.

Finally, our private algorithms obtain accuracy roughly 91%, whereas the same model trained without privacy obtains around 98%. This degradation in accuracy is mostly due to the choice of the optimization algorithm (DP-SGD with clipping); indeed, even without adding any noise, DP-SGD with clipping achieves around 91% accuracy, suggesting that other private optimization algorithms with different clipping strategies (e.g. [34, 6]) may tighten this gap further. As this is not the main focus of our work, we leave this investigation to future work.

## Acknowledgments

HLN is supported by NSF CAREER grant 1750716 and NSF grant 2311649.

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
