# A Nearly unbiased ProjUnit randomizers

Our randomizers in Section 2 and Section 3 may have $O(\log(d)/k)$ bias which can become relatively large when $n \gg d$. In this section, we propose a different normalization technique which allows to provide a sufficiently small bound on the bias, while still enjoying the same guarantees as the fast ProjUnit algorithm. We develop versions of this algorithm for both random rotations (Appendix A.1) and the SRHT transform (Appendix A.2).

## A.1 Unbiased variant of ProjUnit using random rotations

In this section, we describe the modification for random rotation matrices. These transformations are not as efficient as SRHT hence we only present the simple non-correlated version; in the next section we present our unbiased and correlated sampling procedure for SRHT transforms.

For rotationally symmetric distributions of matrices, we slightly modify the algorithm by scaling the output of PrivUnitG by a fixed factor $c$ so that it leads to an unbiased estimate of $v$ i.e. $\mathbb{E}[W^\top \hat{u}] = v$.

---

**Algorithm 5** Unbiased version of ProjUnit using random rotations (client)

---

**Require:** Input vector $v \in \mathbb{R}^d$.
 1: Randomly sample a rotation matrix $W \in \mathbb{R}^{k \times d}$ as described in (1)
 2: Project the input vector $v_p = Wv$
 3: Normalize $u = \frac{v_p}{\|v_p\|_2}$
 4: Let $\hat{u} = c \cdot \text{PrivUnitG}(u)$ where $c = \sqrt{\frac{k}{d}} \frac{\Gamma((d+1)/2)\Gamma(k/2)}{\Gamma((k+1)/2)\Gamma(d/2)}$
 5: Send $\hat{u}$ and (encoding of) $W$ to server

---

We present the details of this modification in Algorithm 5 and state its guarantees in the following theorem. We let $\mathcal{R}_{\mathsf{PU}}$ denote the local randomizer described in Algorithm 5.

**Theorem 5.** *Let* $k \leq d$. *For all unit vectors* $v_1, \ldots, v_n \in \mathbb{R}^d$, *setting* $\hat{\mu} = \mathcal{A}_{\mathsf{PU}}(\mathcal{R}_{\mathsf{PU}}(v_1), \ldots, \mathcal{R}_{\mathsf{PU}}(v_n))$, *the local randomizers* $\mathcal{R}_{\mathsf{PU}}$ *are* $\varepsilon$-DP *and*

$$
\mathbb{E}\left[\left\|\hat{\mu} - \frac{1}{n}\sum_{i=1}^{n} v_i\right\|_2^2\right] \leq \mathsf{Err}_{n,d}(PrivUnitG) \cdot \left(1 + O\left(\frac{\varepsilon + \log k}{k}\right)\right).
$$

*Proof.* The proof proceeds in the same way as Theorem 1. We break the error down into two terms:

$$
\mathbb{E}\left[\left\|\hat{\mu} - \frac{1}{n}\sum_{i=1}^{n} v_i\right\|_2^2\right] = \mathbb{E}\left[\left\|\frac{1}{n}\sum_{i=1}^{n} W_i^\top \hat{u}_i - v_i\right\|_2^2\right]
$$

$$
= \mathbb{E}\left[\left\|\frac{1}{n}\sum_{i=1}^{n} W_i^\top \hat{u}_i - cW_i^\top u_i + cW_i^\top u_i - v_i\right\|_2^2\right]
$$

$$
\stackrel{(i)}{=} \frac{1}{n^2}\sum_{i=1}^{n} \mathbb{E}\left[\left\|W_i^\top \hat{u}_i - cW_i^\top u_i\right\|_2^2\right] + \frac{1}{n^2}\mathbb{E}\left[\left\|\sum_{i=1}^{n} cW_i^\top u_i - v_i\right\|_2^2\right]
$$

$$
\leq \frac{c^2}{n} \max_{i \in [n]} \mathbb{E}\left[\left\|W_i^\top\right\|_2^2\right] \cdot \mathsf{Err}_{1,k}(PrivUnitG) + \frac{1}{n^2}\mathbb{E}\left[\left\|\sum_{i=1}^{n} cW_i^\top u_i - v_i\right\|_2^2\right].
$$

where $(i)$ follows from the fact that PrivUnitG is unbiased and $\mathbb{E}[\hat{u}_i] = cu_i$. The first term is bounded in the same way as before noting that $c = 1 + O(1/k)$ (see Lemma F.1). To analyze the second term, we first show that $\mathbb{E}[cW_i^\top u_i] = v_i$ using a change of variables. Let $W_i' = W_i P_i^\top$ where $P_i$ is the rotation matrix such that $P_i v_i = e_1$, the first standard basis vector. Due to the rotational symmetry of the uniform distribution over rotation matrices, $W_i'$ is also a random rotation matrix. Note that $W_i = W_i' P_i$, hence

$$\mathbb{E}_{W_i}[cW_i^\top u_i] = \mathbb{E}_{W_i'}\left[\frac{c}{\|W_i'P_iv_i\|_2}P_i^\top W_i'^\top W_i'P_iv_i\right]$$

$$= cP_i^\top \mathbb{E}\left[\underbrace{\frac{1}{\|W_i'e_1\|_2}W_i'^\top W_i'e_1}_{z}\right]$$

Notice that $z_j = \frac{1}{\|W_i'e_1\|_2}e_j^\top W_i'^\top W_i'e_1 = \langle W_i'e_j, \frac{1}{\|W_i'e_1\|_2}W_i'e_1\rangle$. Because $W_i'$ is a random rotation matrix (re-scaled by $\sqrt{d/k}$), Lemma F.1 implies that $\mathbb{E}[z_1] = \sqrt{\frac{d}{k}}\frac{\Gamma((k+1)/2)\Gamma(d/2)}{\Gamma((d+1)/2)\Gamma(k/2)} = \frac{1}{c} = 1 + O(1/k)$ and $\mathbb{E}[z_j] = 0$ for all $j > 1$. Thus, $\mathbb{E}[z] = \frac{1}{c}e_1$ and $\mathbb{E}[cW_i^\top u_i] = P_i^\top e_1 = P_i^\top P_iv_i = v_i$. Because $cW_i^\top u_i$ is an unbiased estimator of $v_i$, we have

$$\mathbb{E}\left[\left\|\sum_{i=1}^n cW_i^\top u_i - v_i\right\|_2^2\right] = \sum_{i=1}^n \mathbb{E}\left[\|cW_i^\top u_i - v_i\|_2^2\right]$$

$$= \sum_{i=1}^n \mathbb{E}\left[\|cW_i^\top u_i\|_2^2 + \|v_i\|_2^2 - 2cv_i^\top W_i^\top u_i\right]$$

$$= \sum_{i=1}^n \mathbb{E}\left[\|cW_i^\top u_i\|_2^2 + \|v_i\|_2^2 - 2c\|W_i^\top v_i\|_2\right]$$

$$\le n + \sum_{i=1}^n c^2 \mathbb{E}\left[\|W_i^\top\|_2^2\right]$$

$$\le O(nd/k).$$

Combining all of these together, the claim follows by noting that $\mathsf{Err}_{1,d}(\mathsf{PrivUnitG})/n = \mathsf{Err}_{n,d}(\mathsf{PrivUnitG}) = \Theta(d/n\varepsilon)$. $\qquad\square$

## A.2 Nearly unbiased SRHT-based randomizers

While rescaling by a constant was sufficient to debias the random rotation based randomizer, it is not clear whether such rescaling can debias the SRHT ProjUnit randomizer as it is not rotationally symmetric. To address this, we propose a different normalization technique for the SRHT randomizer which allows to provide tighter upper bounds on the bias. We provide the details for our new client and server protocols in Algorithm 6 and Algorithm 7, respectively.

Let $\mathcal{R}_{\mathsf{UPU}}$ denote the unbiased ProjUnit local randomizer of the client (Algorithm 6), and $\mathcal{A}_{\mathsf{UPU}}$ denote the server aggregation of unbiased ProjUnit (Algorithm 7). We have the following guarantees for this procedure.

**Theorem 6.** *Let* $k \le d$ *and* $\delta = k/n^2d$. *Assume* $k \ge \max\{\varepsilon + \log k, \log^2(nd)\}$. *Then for all unit vectors* $v_1, \ldots, v_n \in \mathbb{R}^d$, *setting* $\hat{\mu} = \mathcal{A}_{\mathsf{UPU}}(\mathcal{R}_{\mathsf{UPU}}(v_1), \ldots, \mathcal{R}_{\mathsf{UPU}}(v_n))$, *the local randomizers* $\mathcal{R}_{\mathsf{UPU}}$ *are* $\varepsilon$-*DP and*

$$\mathbb{E}\left[\left\|\hat{\mu} - \frac{1}{n}\sum_{i=1}^n v_i\right\|_2^2\right] \le \mathsf{Err}_{n,d}(\mathit{PrivUnitG}) \cdot \left(1 + O\left(\frac{\varepsilon + \log k}{k} + \sqrt{\frac{\log^2(nd)}{k}}\right)\right).$$

---

**Algorithm 6** Nearly Unbiased ProjUnit (client)

---

**Require:** Input vector $v \in \mathbb{R}^d$, Bias bound probability $\delta$.

1: Randomly sample diagonal $D$ from the Rademacher distribution based on predefined seed
2: Sample $S \in \mathbb{R}^{k \times d}$ where each row is chosen uniformly at random without replacement from standard basis vectors $\{e_1, \ldots, e_d\}$
3: Let $W = \sqrt{d/k} S H D$
4: Set $C = 1 + \Theta(\sqrt{\log^2(k/\delta)/k})$
5: Project the input vector $v_p = W v$
6: Complete the norm then normalize:

$$
u = \begin{cases} \frac{1}{\sqrt{C}} \left( v_p, \sqrt{C - \|v_p\|_2^2} \right), & \text{if } \|v_p\|_2^2 \leq C \\ \left( \frac{v_p}{\|v_p\|_2}, 0 \right), & \text{otherwise} \end{cases}
$$

7: Let $\hat{u} = \text{PrivUnitG}(u)$
8: Send $C, \hat{u}$ and (encoding of) $S$ to server

---

---

**Algorithm 7** Nearly Unbiased ProjUnit (server)

---

1: Receive $C, \hat{u}_1, \ldots, \hat{u}_1$, from clients with (encodings of) transforms $S_1, \ldots, S_n$
2: Sample the diagonal matrices $D$ from the Rademacher distribution based on predefined seed
3: Let $U = HD$ for where $H \in \mathbb{R}^{d \times d}$ is the Hadamard matrix
4: Return the estimate

$$
\hat{\mu} = \frac{\sqrt{C}}{n} U^\top \sum_{i=1}^n S_i^\top \hat{u}_i[1:k]
$$

---

*Proof.* Note that $\hat{\mu} = \frac{\sqrt{C}}{n} \sum_{i=1}^n W_i^\top \hat{u}_i[1:k]$. Thus we get

$$
\mathbb{E}\left[ \left\| \hat{\mu} - \frac{1}{n} \sum_{i=1}^n v_i \right\|_2^2 \right] \tag{4}
$$

$$
= \mathbb{E}\left[ \left\| \frac{\sqrt{C}}{n} \sum_{i=1}^n W_i^\top \hat{u}_i[1:k] - v_i \right\|_2^2 \right]
$$

$$
= \frac{1}{n} \max_{i \in [n]} \mathbb{E}\left[ \left\| \sqrt{C} W_i^\top \hat{u}_i[1:k] - v_i \right\|_2^2 \right] + \frac{1}{n^2} \sum_{i \neq j} \mathbb{E}\langle \sqrt{C} W_i^\top \hat{u}_i[1:k] - v_i, \sqrt{C} W_j^\top \hat{u}_j[1:k] - v_j \rangle \tag{5}
$$

Now we upper bound both terms in (5) separately. Note that Corollary F.1 implies that with probability at least $1 - n\delta$ we have $\|W_i v_i\|_2^2 \leq (1 + C_1 \sqrt{\log^2(k/\delta)/k}) \|v_i\|_2$ for all $i \in [n]$. We let $E$ denote the event that this event holds. Note that $P(E) \geq 1 - \bar{\delta}$ where $\bar{\delta} = n\delta$.

We begin with the second term in (5). Let $C = 1 + C_1 \sqrt{\log^2(k/\delta)/k}$ for appropriate constant $C_1 > 0$. Note that $W_i = \sqrt{d/k} S_i H D$, taking expecations over the randomness of the local randomizer, and noticing that PrivUnitG is unbiased, we have that for $i \neq j$

$$
\mathbb{E}\langle \sqrt{C} W_i^\top \hat{u}_i[1:k] - v_i, \sqrt{C} W_j^\top \hat{u}_j[1:k] - v_j \rangle = \mathbb{E}\langle \sqrt{C} W_i^\top u_i[1:k] - v_i, \sqrt{C} W_j^\top u_j[1:k] - v_j \rangle \tag{6}
$$

Since $W_i = \sqrt{d/k}S_i HD$ and $W_j = \sqrt{d/k}S_j HD$, we have

$$\mathbb{E}\left[\langle \sqrt{C}W_i^\top u_i[1:k] - v_i, \sqrt{C}W_j^\top u_j[1:k] - v_j\rangle\right]$$

$$= \mathbb{E}\left[\langle W_i^\top W_i v_i - v_i, W_j^\top W_j v_j - v_j\rangle \mid E\right]P(E)$$

$$+ \mathbb{E}\left[\langle \sqrt{C}W_i^\top W_i v_i/\|W_i v_i\|_2 - v_i, \sqrt{C}W_j^\top W_j v_j/\|W_j v_j\|_2 - v_j\rangle \mid E^c\right]P(E^c)$$

$$\leq v_i^T \mathbb{E}[(W_i^\top W_i - I)(W_j^\top W_j - I) \mid E]v_j + \bar{\delta}(2Cd/k + 2)$$

$$\leq v_i^T \mathbb{E}[DH(d/kS_i^\top S_i - I)(d/kS_j^\top S_j - I)HD \mid E]v_j + \bar{\delta}(2Cd/k + 2)$$

$$\overset{(i)}{\leq} \left\|\mathbb{E}[d/kS_i^\top S_i - I \mid E]\right\|_2 \left\|\mathbb{E}[d/kS_j^\top S_j - I \mid E]\right\|_2 + \bar{\delta}(2Cd/k + 2)$$

$$\leq O((\bar{\delta}d/k)^2) + \bar{\delta}(2Cd/k + 2) \leq O(\bar{\delta}d/k),$$

where inequality $(i)$ follows since $\left\|\mathbb{E}[d/kS_j^\top S_j - I \mid E]\right\|_2 \leq O(\bar{\delta}/k)$ since

$$I = \mathbb{E}[(d/k)S_i^\top S_i] = \mathbb{E}[(d/k)S_i^\top S_i \mid E]\mathbb{P}(E) + (1 - \mathbb{P}(E))\mathbb{E}[(d/k)S_i^\top S_i \mid E^c]$$

which implies that

$$\mathbb{E}[(d/k)S_i^\top S_i \mid E] = \frac{I - (1 - \mathbb{P}(E))\mathbb{E}[(d/k)S_i^\top S_i \mid E^c]}{\mathbb{P}(E)}.$$

Since $\mathbb{P}(E) \geq 1 - \bar{\delta}$ and $\left\|(d/k)S_i^\top S_i\right\|_2 \leq (d/k)$, this shows that $\left\|\left(\mathbb{E}\left[(d/k)S_i^\top S_i \mid E\right] - I\right)v\right\|_2 = O(\bar{\delta}d/k)$.

Now we proceed to analyze the first term in (5). Note that for any $i \in [n]$

$$\mathbb{E}\left[\left\|\sqrt{C}W_i^\top \hat{u}_i[1:k] - v_i\right\|_2^2\right] = \mathbb{E}\left[\left\|\sqrt{C}W_i^\top \hat{u}_i[1:k] - \sqrt{C}W_i^\top u_i[1:k] + \sqrt{C}W_i^\top u_i[1:k] - v_i\right\|_2^2\right]$$

$$\overset{(i)}{=} C\mathbb{E}\left[\left\|W_i^\top \hat{u}_i[1:k] - W_i^\top u_i[1:k]\right\|_2^2\right] + \mathbb{E}\left[\left\|\sqrt{C}W_i^\top u_i[1:k] - v_i\right\|_2^2\right]$$

$$\overset{(ii)}{\leq} C\mathbb{E}\left[\left\|W_i^\top\right\|_2^2\right] \cdot C \cdot \mathsf{Err}_{1,k+1}(\mathsf{PrivUnitG}) + \mathbb{E}\left[\left\|\sqrt{C}W_i^\top u_i[1:k] - v_i\right\|_2^2\right].$$

where $(i)$ follows since $\mathbb{E}[\hat{u}_i] = u_i$ as PrivUnitG is unbiased, and $(ii)$ since PrivUnitG is applied for $k + 1$ dimensional vectors of squared norm $C$, hence its error is $C \cdot \mathsf{Err}_{1,k+1}(\mathsf{PrivUnitG})$. For the first term, as $\|W_i\|_2^2 \leq d/k$ and $C = 1 + C_1\sqrt{\log^2(k/\delta)/k}$, we have:

$$C^2 \left\|W_i^\top\right\|_2^2 \cdot \mathsf{Err}_{1,k+1}(\mathsf{PrivUnitG}) \leq C^2\frac{d}{k}c_{k+1,\varepsilon}\frac{k+1}{\varepsilon}$$

$$= C^2\frac{d}{\varepsilon}c_{d,\varepsilon}\frac{c_{k+1,\varepsilon}}{c_{d,\varepsilon}}(1 + 1/k)$$

$$= C^2\frac{d}{\varepsilon}c_{d,\varepsilon} \cdot \left(1 + O\left(\frac{\varepsilon + \log k}{k}\right)\right)$$

$$= \mathsf{Err}_{1,d}(\mathsf{PrivUnitG}) \cdot \left(1 + O\left(\frac{\varepsilon + \log k}{k} + \sqrt{\frac{\log^2(k/\delta)}{k}}\right)\right),$$

where the third step follows from Proposition 5. For the second term, we have

$$\mathbb{E}[\left\|\sqrt{C}W_i^\top u_i[1:k] - v_i\right\|_2^2] = \mathbb{E}\left[\left\|\sqrt{C}W_i^\top u_i[1:k] - \sqrt{C}W_i^\top W_i v_i + \sqrt{C}W_i^\top W_i v_i - v_i\right\|_2^2\right]$$

$$\leq 2C\mathbb{E}\left[\left\|W_i^\top u_i[1:k] - W_i^\top W_i v_i\right\|_2^2\right] + 2\mathbb{E}\left[\left\|\sqrt{C}W_i^\top W_i v_i - v_i\right\|_2^2\right]$$

$$\leq 2C\mathbb{E}\left[\left\|W_i^\top u_i[1:k] - W_i^\top W_i v_i\right\|_2^2\right] + 2\mathbb{E}\left[\left\|\sqrt{C}W_i^\top W_i v_i - W_i^\top W_i v_i\right\|_2^2\right]$$

$$+ 2\mathbb{E}\left[\left\|W_i^\top W_i v_i - v_i\right\|_2^2\right]$$

$$\leq 2C\mathbb{E}\left[\left\|W_i^\top u_i[1:k] - W_i^\top W_i v_i\right\|_2^2\right] + 2(\sqrt{C} - 1)^2 d/k + 2(d/k - 1),$$

where the second inequality follows since $\mathbb{E}[W_i^\top W_i] = I$. Now we have

$$
\begin{aligned}
\mathbb{E}\left[\left\|W_i^\top u_i[1:k] - W_i^\top W_i v_i\right\|_2^2\right] &= \mathbb{E}\left[\left\|W_i^\top W_i v_i/\sqrt{C} - W_i^\top W_i v_i\right\|_2^2 \mid E\right]\mathbb{P}(E) \\
&\quad + \mathbb{E}\left[\left\|W_i^\top W_i v_i/\|W_i v_i\|_2 - W_i^\top W_i v_i\right\|_2^2 \mid E^c\right]\mathbb{P}(E^c) \\
&\leq (1/\sqrt{C}-1)^2 d/k + 2\bar{\delta}(d/k + (d/k)^2) \\
&\leq O\left(\frac{d\sqrt{\log^2(k/\delta)}}{k^{3/2}} + \frac{\bar{\delta}d^2}{k^2}\right).
\end{aligned}
$$

Overall, putting these back in Inequality (5), we get

$$
\begin{aligned}
\mathbb{E}\left[\left\|\hat{\mu} - \frac{1}{n}\sum_{i=1}^n v_i\right\|_2^2\right] &= \frac{1}{n}\mathsf{Err}_{1,d}(\mathsf{PrivUnitG})\cdot\left(1 + O\left(\frac{\varepsilon + \log k}{k} + \sqrt{\frac{\log^2(k/\delta)}{k}}\right)\right) \\
&\quad + O\left(\frac{1}{n}\left(\frac{d\sqrt{\log^2(k/\delta)}}{k^{3/2}} + \frac{n\delta d^2}{k^2} + \frac{d}{k}\right) + (n\delta d/k)^2\right).
\end{aligned}
$$

Noting that $\mathsf{Err}_{1,d}(\mathsf{PrivUnitG})/n = \mathsf{Err}_{n,d}(\mathsf{PrivUnitG}) = c_{d,\varepsilon}\cdot\frac{d}{n\varepsilon}$ for some constant $c_{d,\varepsilon}$, this implies the theorem given that $\delta = k/n^2 d$.

$\square$

## B  Missing proofs for Section 2

### B.1  Proof of Theorem 1

First, note that the claim about privacy follows directly from the privacy guarantees of PrivUnitG [7] as our algorithm applies PrivUnitG over a certain input vector with unit norm.
For accuracy, note that $\hat{\mu} = \frac{1}{n}\sum_{i=1}^n W_i^\top \hat{u}_i$, therefore

$$
\begin{aligned}
\mathbb{E}\left[\left\|\hat{\mu} - \frac{1}{n}\sum_{i=1}^n v_i\right\|_2^2\right] &= \mathbb{E}\left[\left\|\frac{1}{n}\sum_{i=1}^n W_i^\top \hat{u}_i - v_i\right\|_2^2\right] \\
&= \mathbb{E}\left[\left\|\frac{1}{n}\sum_{i=1}^n W_i^\top \hat{u}_i - W_i^\top u_i + W_i^\top u_i - v_i\right\|_2^2\right] \\
&\stackrel{(i)}{=} \frac{1}{n^2}\sum_{i=1}^n \mathbb{E}\left[\left\|W_i^\top \hat{u}_i - W_i^\top u_i\right\|_2^2\right] + \frac{1}{n^2}\mathbb{E}\left[\left\|\sum_{i=1}^n W_i^\top u_i - v_i\right\|_2^2\right] \\
&\leq \frac{1}{n}\max_{i\in[n]}\mathbb{E}\left[\left\|W_i^\top\right\|_2^2\right]\cdot\mathsf{Err}_{1,k}(\mathsf{PrivUnitG}) + \frac{1}{n^2}\mathbb{E}\left[\left\|\sum_{i=1}^n W_i^\top u_i - v_i\right\|_2^2\right].
\end{aligned}
$$

where $(i)$ follows since $\mathbb{E}[\hat{u} \mid W_i = w_i] = u$ as PrivUnitG is unbiased. Now we analyze each of these two terms separately. For the first term, as $\mathbb{E}[\|W_i\|^2] \leq d/k + \beta_{\mathcal{W}}$ for all $i \in [n]$ we have that

is is bounded by

$$\max_{i\in[n]} \mathbb{E}\left[\left\|W^\top\right\|_2^2\right] \cdot \mathsf{Err}_{1,k}(\mathsf{PrivUnitG}) \leq \left(\frac{d}{k}+\beta_{\mathcal{W}}\right)c_{k,\varepsilon}\frac{k}{\varepsilon}$$

$$= \left(\frac{d}{\varepsilon}+\frac{\beta_{\mathcal{W}}k}{\varepsilon}\right)c_{d,\varepsilon}\frac{c_{k,\varepsilon}}{c_{d,\varepsilon}}$$

$$= \left(\frac{d}{\varepsilon}+\frac{\beta_{\mathcal{W}}k}{\varepsilon}\right)c_{d,\varepsilon}\cdot\left(1+O\left(\frac{\varepsilon+\log k}{k}\right)\right)$$

$$= \mathsf{Err}_{1,d}(\mathsf{PrivUnitG})\cdot\left(1+\frac{\beta_{\mathcal{W}}k}{d}+O\left(\frac{\varepsilon+\log k}{k}\right)\right),$$

where the third step follows from Proposition 5. For the second term,

$$\mathbb{E}\left[\left\|\sum_{i=1}^n W_i^\top u_i - v_i\right\|_2^2\right] = \sum_{i=1}^n\sum_{j\neq i}\mathbb{E}\left[\langle W_i^\top u_i - v_i, W_j^\top u_j - v_j\rangle\right] + \sum_{i=1}^n\mathbb{E}\left[\left\|W_i^\top u_i - v_i\right\|_2^2\right]$$

$$\leq \sum_{i=1}^n\sum_{j\neq i}\left\|\mathbb{E}W_i^\top u_i - v_i\right\|_2 \cdot \left\|\mathbb{E}W_j^\top u_j - v_j\right\|_2 + \sum_{i=1}^n\mathbb{E}\left[\left\|W_i^\top u_i - v_i\right\|_2^2\right]$$

$$\leq n(n-1)\alpha_{\mathcal{W}} + \sum_{i=1}^n\mathbb{E}\left[\left\|W_i^\top u_i\right\|_2^2 + \|v_i\|_2^2 - 2v_i^\top W_i^\top u_i\right]$$

$$= n(n-1)\alpha_{\mathcal{W}} + \sum_{i=1}^n\mathbb{E}\left[\left\|W_i^\top u_i\right\|_2^2 + 1 - 2\left\|W_i v_i\right\|_2\right]$$

$$\leq n(n-1)\alpha_{\mathcal{W}} + n\max_{i\in[n]}\mathbb{E}\left[\left\|W_i^\top\right\|_2^2\right] + n.$$

Overall, this shows that

$$\mathbb{E}\left[\left\|\hat{\mu}-\frac{1}{n}\sum_{i=1}^n v_i\right\|_2^2\right] \leq \mathsf{Err}(\mathsf{PrivUnitG}_d, n)\cdot\left(1+O\left(\frac{\varepsilon+\log k}{k}\right)\right)$$

$$+ O\left(\frac{d}{nk}\right) + \frac{1}{n} + \frac{(n-1)\alpha_{\mathcal{W}}}{n}.$$

Noticing that $\mathsf{Err}_{n,d}(\mathsf{PrivUnitG}) = c_{d,\varepsilon}\cdot\frac{d}{n\varepsilon}$ for some constant $c_{d,\varepsilon}$ (see [7]), this implies that

$$\mathbb{E}\left[\left\|\hat{\mu}-\frac{1}{n}\sum_{i=1}^n v_i\right\|_2^2\right] \leq \mathsf{Err}_{n,d}(\mathsf{PrivUnitG})\cdot\left(1+O\left(\frac{\varepsilon+\log k}{k}\right)\right) + \alpha_{\mathcal{W}}.$$

This completes the proof.

### B.2 Proof of Proposition 1

The first item follows immediately as $U\in\mathbb{R}^{d\times d}$ is a random rotation matrix where $U^\top U = I$, hence $\|U\|\leq 1$.

For the second item, we use a change of variables. Let $W' = WP^\top$ where $P$ is the rotation matrix such that $Pv = e_1$, the first standard basis vector. Recall that the rotation matrix $P$ is orthogonal i.e. $P^\top = P^{-1}$. Due to the rotational symmetry of rotation matrices, $W'$ is a random also a random rotation matrix. Note that $W = W'P$.

$$\mathbb{E}_W\left[\frac{W^\top W v}{\|Wv\|_2}\right] = \mathbb{E}_{W'}\left[\frac{1}{\|W'Pv\|_2}P^\top W'^\top W'Pv\right]$$

$$= P^\top\mathbb{E}\left[\underbrace{\frac{1}{\|W'e_1\|_2}W'^\top W'e_1}_{z}\right]$$

Notice that $z_j = \frac{1}{\|W'e_1\|_2} e_j^\top W'^\top W' e_1 = \langle W'e_j, \frac{1}{\|W'e_1\|_2} W'e_1 \rangle$. First, note that $\mathbb{E}[z_j] = 0$ for all $j > 1$. Moreover, $z_1 = \|W'e_1\|_2$, therefore because $W'$ is a random rotation matrix, Lemma F.1 implies that $\mathbb{E}[z_1] = \sqrt{d/k}(\sqrt{k/d} + O(1/\sqrt{kd})) = 1 + O(1/k)$. We let $c = \mathbb{E}[z_1]$. Thus, $\mathbb{E}[z] = ce_1$ and $\mathbb{E}[W^\top Wv/\|Wv\|_2] = cP^\top e_1 = cP^\top Pv = cv$. Therefore, $\left\|\mathbb{E}[\frac{W^\top Wv}{\|Wv\|_2} - v]\right\|_2 = |c - 1|\|v\|_2 = O(1/k)$.

## B.3 Proof of Proposition 2

The bound on the operator norm is straightforward and follows from the fact that the Hadamard transform has operator norm bounded by 1.

Next we bound the bias. Let $\delta = \min(1/d^2, k/2d)$ and let $E_1$ denote the event where $\|Wv\| \in 1 \pm O(\ln(k/\delta)/\sqrt{k})$. By Corollary F.1, $E_1$ happens with probability $1 - \delta$. Note that $W^\top W$ is PSD and $\mathbb{E}[W^\top W] = I$. Thus,

$$
\begin{aligned}
\left\|\mathbb{E}\left[\frac{W^\top Wv}{\|Wv\|} - W^\top Wv\right]\right\| &\leq \left\|\mathbb{E}\left[\frac{W^\top Wv}{\|Wv\|} - W^\top Wv | E_1\right]\right\| + \left\|\mathbb{E}\left[\frac{W^\top Wv}{\|Wv\|} - W^\top Wv | \overline{E_1}\right]\right\| \mathbb{P}(\overline{E_1}) \\
&\leq \left\|\mathbb{E}\left[\left(\frac{1}{\|Wv\|} - 1\right) W^\top Wv | E_1\right]\right\| + (\|W^\top\| + 1)\mathbb{P}(\overline{E_1}) \\
&\leq \left\|\mathbb{E}\left[\left(\frac{1}{\|Wv\|} - 1\right) W^\top W | E_1\right]\right\| + (\|W^\top\| + 1)\mathbb{P}(\overline{E_1}) \\
&\leq \left\|\mathbb{E}\left[\left|\frac{1}{\|Wv\|} - 1\right| W^\top W | E_1\right]\right\| + (\|W^\top\| + 1)\mathbb{P}(\overline{E_1}) \\
&\leq O(\ln(k/\delta)/\sqrt{k}) + (\sqrt{d/k} + 1)\delta
\end{aligned}
$$

Substituting in the value of $\delta$ gives the desired bound on the bias, by noticing that $\left\|\mathbb{E}[W^\top W \mid E_1]\right\|_2 \leq 2$ since for any unit vector $x$,

$$
\begin{aligned}
x^\top I x &= \mathbb{E}[x^\top W^\top W x] \\
&= \mathbb{E}[x^\top W^\top W x \mid E_1]\mathbb{P}(E_1) + \mathbb{E}[x^\top W^\top W x \mid \overline{E_1}]\mathbb{P}(\overline{E_1}) \\
&\geq \mathbb{E}[x W^\top W x \mid E_1](1 - \delta).
\end{aligned}
$$

In other words, $\mathbb{E}[x W^\top W x \mid E_1] \leq \frac{1}{1-\delta}$ for all $x \in \mathbb{R}^d$ with unit norm.

## C  Proof of Theorem 4

The proof of this result follows from the next proposition.

**Proposition 3.** *Let* $k \leq d$, $G \geq 1$ *be an integer,* $U_1, \ldots, U_G$ *and* $W_1, \ldots, W_n$ *be sampled as described in* (3)*. Moreover,* $U_j$ *for* $j \in [G]$ *and* $W_i$ *for* $i \in [n]$ *satisfy:*

    1. *Bounded operator norm:* $\left\|U_j^\top\right\| \leq 1$.

    2. *Bounded bias:* $\left\|\mathbb{E}\left[\frac{W_i^\top W_i v}{\|W_i v\|_2}\right] - v\right\|_2 \leq \sqrt{\alpha_\mathcal{W}}$ *for all unit vectors* $v \in \mathbb{R}^d$.

*Then for all unit vectors* $v_1, \ldots, v_n \in \mathbb{R}^d$, *setting* $\hat{\mu} = \mathcal{A}_{\mathsf{CPU}}(\mathcal{R}_{\mathsf{CPU}}(v_1), \ldots, \mathcal{R}_{\mathsf{CPU}}(v_n))$,

$$
\mathbb{E}\left[\left\|\hat{\mu} - \frac{1}{n}\sum_{i=1}^n v_i\right\|_2^2\right] \leq \mathsf{Err}_{n,d}(\mathit{PrivUnitG}) \cdot \left(1 + O\left(\frac{\varepsilon + \log k}{k} + \frac{n\varepsilon \log^2(nd)}{Gdk}\right)\right) + \alpha_\mathcal{W}.
$$

Before proving the proposition, we can now prove Theorem 4.

*Proof.* The proof follows from Proposition 3 by noting that the server returns $\sum_{i=1}^n W_i^\top \hat{u}_i/n$. The first property holds immediately from the definition of $U_j$. Moreover, for the second property,

Proposition 2 implies that $\alpha_{\mathcal{W}} = O(\log^2(d)/k)$. Since $\mathsf{Err}_{n,d}(\mathsf{PrivUnitG}) = \Theta(d/n\varepsilon)$, the claim about utility follows.

Now we prove the part regarding runtime. First, note that calculating the matrix $D$ can be done efficiently using standard techniques [36, 5]. The server has to calculate the quantity

$$U^\top \sum_{i=1}^n S_i^\top \hat{u}_i.$$

Note that the summation has vectors which are $k$-sparse, therefore can be done in time $O(nk)$. Then, we have a multiplication step by Hadamard transform, which can be done in $O(d\log d)$. $\qquad\square$

We now prove Proposition 3.

*Proof.* Note that $\hat{\mu} = \frac{1}{n}\sum_{i=1}^n W_i^\top \hat{u}_i$. Therefore we have

$$\mathbb{E}\left[\left\|\hat{\mu} - \frac{1}{n}\sum_{i=1}^n v_i\right\|_2^2\right] = \mathbb{E}\left[\left\|\frac{1}{n}\sum_{i=1}^n W_i^\top \hat{u}_i - v_i\right\|_2^2\right]$$

$$= \mathbb{E}\left[\left\|\frac{1}{n}\sum_{i=1}^n W_i^\top \hat{u}_i - W_i^\top u_i + W_i^\top u_i - v_i\right\|_2^2\right]$$

$$\overset{(i)}{=} \frac{1}{n^2}\sum_{i=1}^n \mathbb{E}\left[\left\|W_i^\top \hat{u}_i - W_i^\top u_i\right\|_2^2\right] + \frac{1}{n^2}\mathbb{E}\left[\left\|\sum_{i=1}^n W_i^\top u_i - v_i\right\|_2^2\right]$$

$$\leq \frac{1}{n}\max_{i\in[n]}\mathbb{E}\left[\left\|W_i^\top\right\|_2^2\right]\cdot \mathsf{Err}_{1,k}(\mathsf{PrivUnitG}) + \frac{1}{n^2}\mathbb{E}\left[\left\|\sum_{i=1}^n W_i^\top u_i - v_i\right\|_2^2\right].$$

where $(i)$ follows since $\mathbb{E}[\hat{u}] = u$ as PrivUnitG is unbiased. Now we analyze each of these two terms separately. For the first term, as $\mathbb{E}[\|W_i\|^2] \leq d/k$ for all $i\in[n]$ we have that it is bounded by

$$\max_{i\in[n]}\mathbb{E}\left[\left\|W_i^\top\right\|_2^2\right]\cdot\mathsf{Err}_{1,k}(\mathsf{PrivUnitG}) \leq \frac{d}{k}c_{k,\varepsilon}\frac{k}{\varepsilon} = \frac{d}{\varepsilon}c_{d,\varepsilon}\frac{c_{k,\varepsilon}}{c_{d,\varepsilon}}$$

$$= \frac{d}{\varepsilon}c_{d,\varepsilon}\cdot\left(1 + O\left(\frac{\varepsilon + \log k}{k}\right)\right)$$

$$= \mathsf{Err}_{1,d}(\mathsf{PrivUnitG})\cdot\left(1 + O\left(\frac{\varepsilon + \log k}{k}\right)\right),$$

where the third step follows from Proposition 5. For the second term,

$$\mathbb{E}\left[\left\|\sum_{i=1}^n W_i^\top u_i - v_i\right\|_2^2\right] = \sum_{i=1}^n\sum_{j\neq i}\mathbb{E}\left[\langle W_i^\top u_i - v_i, W_j^\top u_j - v_j\rangle\right] + \sum_{i=1}^n\mathbb{E}\left[\left\|W_i^\top u_i - v_i\right\|_2^2\right].$$

For the second term note that

$$\sum_{i=1}^n\mathbb{E}\left[\left\|W_i^\top u_i - v_i\right\|_2^2\right] = \sum_{i=1}^n\mathbb{E}\left[\left\|W_i^\top u_i\right\|_2^2 + \|v_i\|_2^2 - 2v_i^\top W_i^\top u_i\right]$$

$$= \sum_{i=1}^n\mathbb{E}\left[\left\|W_i^\top u_i\right\|_2^2 + 1 - 2\|W_i v_i\|_2\right]$$

$$\leq n\max_{i\in[n]}\mathbb{E}\left[\left\|W_i^\top\right\|_2^2\right] + n \leq n(d/k + 1).$$

For the first term, we have

$$\mathbb{E}\left[\langle W_i^\top u_i - v_i, W_j^\top u_j - v_j\rangle\right]$$

$$= \mathbb{E}\left[\langle W_i^\top(u_i - Wv_i) + W_i^\top W_i v_i - v_i, W_j^\top(u_j - W_j v_j) + W_j^\top W_j v_j - v_j\rangle\right]$$

$$= \mathbb{E}\left[\langle W_i^\top(u_i - Wv_i), W_j^\top(u_j - W_j v_j)\rangle\right] + \mathbb{E}\left[\langle W_i^\top W_i v_i - v_i, W_j^\top(u_j - W_j v_j)\rangle\right]$$

$$\quad + \mathbb{E}\left[\langle W_i^\top(u_i - Wv_i), W_j^\top W_j v_j - v_j\rangle\right] + \mathbb{E}\left[\langle W_i^\top W_i v_i - v_i, W_j^\top W_j v_j - v_j\rangle\right]$$

Because $\mathbb{E}_{S_i}\left[\frac{d}{k}S_i^\top S_i\right] = I$, $H^\top H = I$, and $D^\top D = I$, we can evaluate the second term:

$$\mathbb{E}_{S_i}\left[\langle W_i^\top W_i v_i - v_i, W_j^\top (u_j - W_j v_j)\rangle\right] = \mathbb{E}_{S_i}\left[\left\langle \frac{d}{k}D^\top H^\top S_i^\top S_i H D v_i - v_i, W_j^\top (u_j - W_j v_j)\right\rangle\right] = 0$$

Similarly, we can evaluate the fourth term:

$$\mathbb{E}\left[\langle W_i^\top W_i v_i - v_i, W_j^\top W_j v_j - v_j\rangle\right] = \mathbb{E}\left[\left\langle \frac{d}{k}D^\top H^\top S_i^\top S_i H D v_i - v_i, W_j^\top W_j v_j - v_j\right\rangle\right] = 0$$

The third term is similar. Thus, we only need to bound the first term. First we give an upper bound that holds with probability 1.

$$\langle W_i^\top (u_i - W_i v_i), W_j^\top (u_j - W_j v_j)\rangle \le \left\|W_i^\top\right\|\left\|W_j^\top\right\|(\|W_i\| + 1)(\|W_j\| + 1)$$
$$\le O\left(d^2/k^2\right)$$

Let $E_1$ be the event that $\|W_i v_i\|, \|W_j v_j\| \in 1 \pm O\left(\ln(k/\delta)/\sqrt{k}\right)$ where $\delta$ is a parameter to be chosen later. We will split the expectation depending on the event $E_1$.

$$\langle W_i^\top (u_i - W_i v_i), W_j^\top (u_j - W_j v_j)\rangle$$
$$= \left(\frac{1}{\|W_i v_i\|} - 1\right)\left(\frac{1}{\|W_j v_j\|} - 1\right) v_i^\top W_i^\top W_i W_j^\top W_j v_j$$
$$= \frac{d^2}{k^2}\left(\frac{1}{\|W_i v_i\|} - 1\right)\left(\frac{1}{\|W_j v_j\|} - 1\right) v_i^\top U^\top S_i^\top S_i \underbrace{U U^\top}_{I} S_j^\top S_j U v_j$$

Note that $M := S_i^\top S_i S_j^\top S_j$ is PSD (both $S_i^\top S_i$ and $S_j^\top S_j$ are diagonal matrices and so is the product). Furthermore, $\mathbb{E}[\frac{d}{k}S_i^\top S_i] = I$. Thus

$$\mathbb{E}\left[\frac{d^2}{k^2}\left(\frac{1}{\|W_i v_i\|} - 1\right)\left(\frac{1}{\|W_j v_j\|} - 1\right) v_i^\top U^\top M U v_j, E_1\right] \cdot \Pr[E_1]$$
$$\le \mathbb{E}\left[\frac{d^2}{4k^2}\left(\frac{1}{\|W_i v_i\|} - 1\right)\left(\frac{1}{\|W_j v_j\|} - 1\right)(v_i^\top + v_j^\top)U^\top M U (v_i + v_j), E_1\right] \cdot \Pr[E_1]$$
$$\le \mathbb{E}\left[O\left(\ln^2(k/\delta)d^2/k^3\right)(v_i^\top + v_j^\top)U^\top M U (v_i + v_j)\right]$$
$$= O\left(\ln^2(k/\delta)/k\right)$$

Therefore,

$$\mathbb{E}\left[\langle W_i^\top (u_i - W_i v_i), W_j^\top (u_j - W_j v_j)\rangle\right] \le O\left(\ln^2(k/\delta)/k\right) + O\left(d^2/k^2\right) \cdot \Pr\left[\overline{E_1}\right]$$

We now complete the proof of the claim. Combining the analysis, we get

$$\mathbb{E}\left[\left\|\hat{\mu} - \frac{1}{n}\sum_{i=1}^n v_i\right\|_2^2\right] \le \mathsf{Err}_{n,d}(\mathsf{PrivUnitG}) \cdot \left(1 + O\left(\frac{\varepsilon + \log k}{k}\right)\right)$$
$$+ O\left(\frac{d}{nk} + \frac{d^2\delta}{Gk^2} + \frac{\ln^2(k/\delta)}{Gk} + \alpha_{\mathcal{W}}\right).$$

Noticing that $\mathsf{Err}_{1,d}(\mathsf{PrivUnitG})/n = \mathsf{Err}_{n,d}(\mathsf{PrivUnitG}) = c_{d,\varepsilon} \cdot \frac{d}{n\varepsilon}$ for some constant $c_{d,\varepsilon}$ (see [7]), and $\delta \le k/(nd)$, this implies that

$$\mathbb{E}\left[\left\|\hat{\mu} - \frac{1}{n}\sum_{i=1}^n v_i\right\|_2^2\right] \le \mathsf{Err}_{n,d}(\mathsf{PrivUnitG}) \cdot \left(1 + O\left(\frac{\varepsilon + \log k}{k} + \frac{n\varepsilon \ln^2(k/\delta)}{Gdk}\right)\right) + \alpha_{\mathcal{W}}.$$

This proves the claim. $\qquad\square$

# D  ProjUnit using Gaussian transforms

Building on the randomized projection framework of the previous section, in this section we instantiate it with the Gaussian transform. In particular, we sample $W \in \mathcal{R}^{k \times d}$ from the Gaussian distribution where $W$ has i.i.d. $\mathcal{N}(0, 1/k)$ entries. The following theorem states our guarantees for this distribution.

**Theorem 7.** *Let $k \leq d$ and $W \in \mathcal{R}^{k \times d}$ be sampled from the Gaussian distribution where $W$ has i.i.d. $\mathcal{N}(0, 1/k)$ entries. Then for all unit vectors $v_1, \ldots, v_n \in \mathbb{R}^d$, setting $\hat{\mu} = \mathcal{A}_{\mathsf{PU}}(\mathcal{R}_{\mathsf{PU}}(v_1), \ldots, \mathcal{R}_{\mathsf{PU}}(v_n))$,*

$$
\mathbb{E}\left[\left\|\hat{\mu} - \frac{1}{n}\sum_{i=1}^{n} v_i\right\|_2^2\right] \leq \mathsf{Err}(PrivUnitG_d, n)\left(1 + O\left(\sqrt{\frac{k}{d}} + \frac{\varepsilon + \log k}{k}\right)\right) + O\left(\frac{1}{k^2}\right).
$$

The proof follows directly from Theorem 1 and the following proposition which proves certain properties of the Guassian transform.

**Proposition 4.** *Consider $W \in \mathbb{R}^{k \times d}$ with i.i.d. $\mathcal{N}(0, 1/k)$ entries and a fixed $v \in \mathbb{R}^d$. Then*

1. *Bounded operator norm:*

$$
\mathbb{E}\|W^\top\|^2 \leq \frac{d}{k}\left(1 + O\left(\sqrt{\frac{k}{d}}\right)\right).
$$

2. *Bounded bias: for every unit vector $v \in \mathbb{R}^d$*

$$
\left\|\mathbb{E}\frac{W^\top W v}{\|W v\|} - v\right\| = O(1/k).
$$

*Proof.* For the first item, we rely on standard results in random matrix theory. If we let Z denote the top singular value of $\sqrt{k}W^\top$, then (17, Theorem 2.13) shows that for any $t$, $\Pr(Z > \sqrt{d} + \sqrt{k} + t) < \exp(-t^2/2)$. This implies that $median(Z) \leq \sqrt{d} + \sqrt{k} + 2$. Further, by the isoperimetric inequality, $Z$ is concentrated around its median with subGaussian tails, so that the second moment of $Z - median(Z)$ is at most $O(1)$. Thus the second moment of $Z$ is at most $median(Z)^2 + O(1) \leq (\sqrt{d} + \sqrt{k} + 2)^2 + O(1)$. Scaling this by $k$, we conclude that $\mathbb{E}[\|W^\top\|_{op}^2] \leq \frac{d}{k}(1 + 2\sqrt{\frac{k}{d}} + \frac{O(1)}{k})$.

For the second item, we use a change of variables. Let $W' = WP^\top$ where $P$ is the rotation matrix such that $Pv = e_1$, the first standard basis vector. Recall that the rotation matrix $P$ is orthogonal i.e. $P^\top = P^{-1}$. Due to the rotational symmetry of the normal distribution, $W'$ is a random matrix with i.i.d. $\mathcal{N}(0, 1/k)$ entries. Note that $W = W'P$.

$$
\mathbb{E}_W[W^\top u] = \mathbb{E}_{W'}\left[\frac{1}{\|W'Pv\|_2}P^\top {W'}^\top W'Pv\right]
$$

$$
= P^\top \mathbb{E}\left[\underbrace{\frac{1}{\|W'e_1\|_2}{W'}^\top W'e_1}_{z}\right]
$$

Notice that $z_j = \frac{1}{\|W'e_1\|_2}e_j^\top {W'}^\top W'e_1 = \langle W'e_j, \frac{1}{\|W'e_1\|_2}W'e_1\rangle$. Because $W'$ has i.i.d. $\mathcal{N}(0, 1/k)$ entries, $z_1 = \|W'e_1\|_2$ is $1/\sqrt{k}$ times a $\chi$ random variable with $k$ degrees of freedom. We let $\frac{1}{c} = \mathbb{E}[z_1] = \frac{1}{\sqrt{k}} \cdot \frac{\sqrt{2}\Gamma((k+1)/2)}{\Gamma(k/2)} = 1 - O(1/k)$ and $\mathbb{E}[z_j] = 0 \; \forall j > 1$. Thus, $\mathbb{E}[z] = \frac{1}{c}e_1$ and $\mathbb{E}[W_i^\top u_i] = \frac{1}{c}P_i^\top e_1 = \frac{1}{c}P^\top Pv = \frac{1}{c}v$. Therefore, $\left\|\mathbb{E}[\frac{W^\top W v}{\|Wv\|_2} - v]\right\|_2 = \|1/c - 1|v\|_2 = O(1/k)$. $\qquad \square$

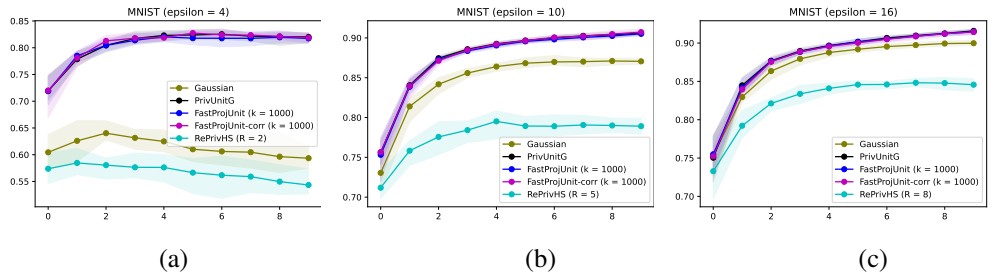

(a)            (b)            (c)

Figure 5: Test accuracy on the MNIST dataset as a function of epoch for (a) $\varepsilon = 4.0$, (b) $\varepsilon = 10.0$ and (c) $\varepsilon = 16.0$.

## E  Additional plots for the MNIST experiment

We present additional details for the MNIST experiment including the description of the neural network (Table 2) and additional plots with different values of the privacy parameters for the MNIST experiment. In Figure 5, we present more plots for the MNIST experiment where we train models with several privacy parameters $\varepsilon \in \{4, 10, 16\}$.

| Layer | Parameters |
|---|---|
| Convolution $+ \tanh$ | 16 filters of $8 \times 8$, stride 2, padding 2 |
| Average pooling | $2 \times 2$, stride 1 |
| Convolution $+ \tanh$ | 32 filters of $4 \times 4$, stride 2, padding 0 |
| Average pooling | $2 \times 2$, stride 1 |
| Fully connected $+ \tanh$ | 32 units |
| Fully connected $+ \tanh$ | 10 units |

Table 2: Architecture for convolutional network model.

## F  Helper Lemmas

### F.1  Helper lemmas for random rotations

**Lemma F.1.** *Let $x$ be a random unit vector on the unit ball of $\mathbb{R}^d$ and $z$ be the projection of $x$ on to the last $k$ coordinates. We have*

$$\left| \mathbb{E}[\|z\|] - \sqrt{k/d} \right| = O\left( \frac{1}{\sqrt{kd}} \right)$$

*Proof.* We represent $d$ dimensional vector $x$ using spherical coordinates as follows.

$$x_1 = \cos(\phi_1)$$
$$x_2 = \sin(\phi_1)\cos(\phi_2)$$
$$\cdots$$
$$x_{d-1} = \sin(\phi_1)\cdots\sin(\phi_{d-2})\cos(\phi_{d-1})$$
$$x_d = \sin(\phi_1)\cdots\sin(\phi_{d-2})\sin(\phi_{d-1})$$

The squared length of the projection is

$$\sum_{i=d-k+1}^{d} x_i^2 = \sin^2(\phi_1)\cdots\sin^2(\phi_{d-k})$$

Recall the surface area element is $\sin^{d-2}(\phi_1)\sin^{d-3}(\phi_2)\cdots\sin(\phi_{d-2})\,d\phi_1\cdots d\phi_{d-1}$.

For $k \geq 2$, the expected length is

$$\frac{\int_0^\pi \cdots \int_0^\pi \int_0^{2\pi} (\sin(\phi_1)\cdots\sin(\phi_{d-k}))\sin^{d-2}(\phi_1)\sin^{d-3}(\phi_2)\cdots\sin(\phi_{d-2})\,d\phi_1\cdots d\phi_{d-1}}{S_{d-1}}$$

where $S_{d-1}$ is the surface area of the unit ball, which is $S_{d-1} = \frac{2\pi^{d/2}}{\Gamma(d/2)}$.

We first evaluate the integral for each sine power.

**Claim F.1.** *For integer $n \geq 1$ we have*

$$\int_0^\pi \sin^n x\,dx = \frac{\Gamma((n+1)/2)}{\Gamma(1+n/2)}\sqrt{\pi}$$

*Proof.* For $n \geq 2$, we have

$$\int \sin^n x\,dx = -\int \sin^{n-1} x\,d(\cos x)$$

$$= -\sin^{n-1} x \cos x + (n-1)\int \sin^{n-2} x \cos^2 x\,dx$$

$$= -\sin^{n-1} x \cos x + (n-1)\int \sin^{n-2} x(1-\sin^2 x)\,dx$$

Thus,

$$\int_0^\pi \sin^n x\,dx = \frac{n-1}{n}\int_0^\pi \sin^{n-2} x\,dx - \frac{\sin^{n-1} x \cos x}{n}\Big|_0^\pi$$

$$= \frac{(n-1)/2}{n/2}\int_0^\pi \sin^{n-2} x\,dx$$

The claim then follows using induction with base cases $\int_0^\pi \sin x\,dx = 2$ and $\int_0^\pi dx = \pi$. $\qquad\square$

$$\int_0^\pi \cdots \int_0^\pi \int_0^{2\pi} (\sin(\phi_1)\cdots\sin(\phi_{d-k}))\sin^{d-2}(\phi_1)\sin^{d-3}(\phi_2)\cdots\sin(\phi_{d-2})\,d\phi_1\cdots d\phi_{d-1}$$

$$=2\pi\int_0^\pi \cdots \int_0^\pi \sin^{d-1}(\phi_1)\cdots\sin^k(\phi_{d-k})\sin^{k-2}(\phi_{d-k+1})\cdots\sin(\phi_{d-2})\,d\phi_1\cdots d\phi_{d-2}$$

$$=2\pi\frac{\Gamma(d/2)}{\Gamma((d+1)/2)}\sqrt{\pi}\cdots\frac{\Gamma((k+1)/2)}{\Gamma((k+2)/2)}\sqrt{\pi}\cdot\frac{\Gamma((k-1)/2)}{\Gamma(k/2)}\sqrt{\pi}\cdots\frac{\Gamma(1)}{\Gamma(3/2)}\sqrt{\pi}$$

$$=2\pi^{d/2}\frac{\Gamma((k+1)/2)}{\Gamma((d+1)/2)\Gamma(k/2)}$$

The expected length is

$$\mathbb{E}[\|z\|] = \frac{\Gamma((k+1)/2)\Gamma(d/2)}{\Gamma((d+1)/2)\Gamma(k/2)}$$

$$= \frac{\Gamma(k)2^d\Gamma(d/2)^2}{2^k\Gamma(k/2)^2\Gamma(d)}$$

$$= \frac{\sqrt{k}\left(1 - \frac{1}{4k} + O\left(1/k^2\right)\right)}{\sqrt{d}\left(1 - \frac{1}{4d} + O\left(1/d^2\right)\right)}$$

In the second line, we use the Legendre duplication formula $\Gamma(k/2)\Gamma((k+1)/2) = 2^{1-k}\sqrt{\pi}\Gamma(k)$.
In the third line, we use the Stirling's approximation $\Gamma(z) = \sqrt{2\pi/z}(z/e)^z(1+1/(12z)+O(1/z^2))$.
$\qquad\square$

## F.2   SRHT Analysis

The Subsampled Randomized Hadamard Transform (SRHT) is the random matrix ensemble defined as $W = \sqrt{\frac{d}{k}}SHD$. Here $D \in \mathbb{R}^{d \times d}$ is a diagonal matrix with independent uniform $\pm 1$ values on its diagonal, and $H \in \mathbb{R}^{d \times d}$ is the normalized Hadamard transform ($H_{i,j} = (-1)^{\langle v(i), v(j) \rangle}/\sqrt{d}$, where $v(i)$ is the $(\log_2 d)$-dimensional vector obtained by writing $i$ in binary). The matrix $S \in \mathbb{R}^{k \times d}$ is a sampling matrix. The fact that the SRHT preserves the Euclidean norm of any fixed vector with large probability has been known for some time [3, 16, 35], though different works have analyzed slightly different variants of the SRHT, all having to do with how $S$ is defined.

In this work, we make use of the SRHT in which $S$ samples without replacement: that is, each row of $S$ has a $1$ in a uniformly random entry and zeroes elsewhere, and no two rows of $S$ are equal. The tightest known analysis of the SRHT [16] analyzes the SRHT with a different sampling matrix: $S_\eta = diag(\eta)$, where $\eta_1, \ldots, \eta_d$ are independent Bernoulli random variables each with expectation $k/d$ (so that we sample a *random* number of rows from $HD$, which is equal to $k$ only in expectation).

The following is a special case of Theorem 9 in the full version of [16]

**Theorem 8** ([16]). *Suppose $W = \sqrt{\frac{d}{k}}S_\eta HD$ for $S = diag(\eta)$, where $\eta_1, \ldots, \eta_d$ is a sequence of independent, uniform Bernoulli random variables each with expectation $k/d$. Then for some constant $C > 0$, for any fixed $u \in \mathbb{R}^d$ of unit Euclidean norm and $\delta \in (0, 1)$,*

$$\Pr_{\eta, D}(|\|Wu\|_2^2 - 1| > C\sqrt{\log(1/\delta)\log(k/\delta)/k}) < \delta$$

An analysis of the SRHT using sampling without replacement then follows as a corollary.

**Corollary F.1.** *Suppose $W = \sqrt{\frac{d}{k}}SHD$ is obtained with $S$ being a $k \times d$ sampling matrix without replacement. Then for some constant $C > 0$, for any fixed $u \in \mathbb{R}^d$ of unit Euclidean norm and $\delta \in (0, 1)$,*

$$\Pr_{\eta, D}(|\|Wu\|_2^2 - 1| > C\sqrt{\log^2(k/\delta)/k}) < \delta$$

*Proof.* Consider $W' = \sqrt{\frac{d}{k}}S_\eta HD$ with Bernoulli parameter $k/d$, as in Theorem 8. Then for any $\delta' \in (0, 1)$ and fixed unit vector $u \in \mathbb{R}^d$, $\Pr_{\eta, D}(E) < \delta'$, where $E$ is the event that $|\|W'u\|_2^2 - 1| > C\sqrt{\log(1/\delta')\log(k/\delta')/k}$. But we also have

$$\begin{aligned}
\Pr(E) &\geq \Pr(E \cap (\|\eta\|_1 = k)) \\
&= \Pr(E \mid \|\eta\|_1 = k) \cdot \Pr(\|\eta\|_1 = k) \\
&= \Pr(E \mid \|\eta\|_1 = k) \cdot \Theta(1/\sqrt{k}).
\end{aligned}$$

Note $\Pr(E \mid \|\eta\|_1 = k)$ is exactly $\Pr(|\|Wu\|_2^2 - 1| > C\sqrt{\log(1/\delta')\log(k/\delta')/k})$, where $W$ is defined by sampling without replacement. Thus we have

$$\Pr(|\|Wu\|_2^2 - 1| > C\sqrt{\log(1/\delta')\log(k/\delta')/k}) < C\delta'\sqrt{k}.$$

The claim then follows by applying the above with $\delta' = \delta/(C\sqrt{k})$. $\square$

# G   Details of PrivUnitG

For completeness, in this section we provide the full details of PrivUnitG which was proposed by Asi et al. [7]. Roughly, this algorithm uses the normal distribution to approximate the uniform distribution over the sphere for large dimensions. We refer the reader to [7] for more details about PrivUnitG.

In the algorithm, $\Phi$ and $\phi$ denote the Cumulative distribution function and probability density function for a Gaussian random variable $\mathsf{N}(0, I_d)$. There are multiple ways to set the parameters of PrivUnitG to achieve $\varepsilon$-DP; in our paper, we use the optimized parameters as described by Asi et al. [7], which allow to minimize the expected mean squared error (see Proposition 4 in [7]).

We note that Algorithm 8 describes the clients' algorithm (local randomizers) in the PrivUnitG protocol. The server aggregation simply adds all messages received from clients. Thus, we let $\mathcal{R}_{\mathrm{PrivUnitG}_\varepsilon}$ denote the local randomizer in Algorithm 8 (with optimized parameters to satisfy $\varepsilon$-DP) and let $\mathcal{A}_{\mathrm{PrivUnitG}_\varepsilon}$ denote the additive server aggregation.

---

**Algorithm 8** PrivUnitG$(p, q)$

---

**Require:** $v \in \mathbb{S}^{d-1}$, $q \in [0, 1]$, $p \in [0, 1]$
1: Draw $z \sim \mathsf{Ber}(p)$
2: Let $U = \mathsf{N}(0, \sigma^2)$ where $\sigma^2 = 1/d$
3: Set $\gamma = \Phi_{\sigma^2}^{-1}(q) = \sigma \cdot \Phi^{-1}(q)$
4: **if** $z = 1$ **then**
5:     Draw $\alpha \sim U \mid U \geq \gamma$
6: **else**
7:     Draw $\alpha \sim U \mid U < \gamma$
8: Draw $V^\perp \sim \mathsf{N}(0, \sigma^2(I - vv^T))$
9: Set $V = \alpha v + V^\perp$
10: Calculate
$$m = \sigma \phi(\gamma/\sigma) \left( \frac{p}{1-q} - \frac{1-p}{q} \right)$$
11: Return $\frac{1}{m} \cdot V$

---

We also use the following useful result on the error of PrivUnitG for different dimensions. Recall that $\mathsf{Err}_{n,d}(\mathrm{PrivUnitG}) = c_{d,\varepsilon} \frac{d}{n\varepsilon}$. Then we have the following.

**Proposition 5** (Propsition 5, [7]). *Fix $\varepsilon > 0$. For any $1 \leq k \leq d$,*

$$1 - O\left( \frac{\varepsilon + \log k}{k} + \frac{\varepsilon}{k} \right) \leq \frac{c_{k,\varepsilon}}{c_{d,\varepsilon}} \leq 1 + O\left( \frac{\varepsilon + \log k}{k} + \frac{\varepsilon}{k} \right).$$

# H   Compressed PrivUnitG

Compressing the PrivUnit (resp. PrivUnitG) algorithm, using the technique of Feldman and Talwar [24], requires a pseudorandom generator that generates samples from a unit ball (resp. Gaussian) and fools spherical caps. As observed in [24], such PRGs with small seed length are known [30, 28]. However, the constructions in those works are optimized for seed length, and the computational cost of expanding a seed to a vector is a large polynomial. In this section, we argue that for inputs $x$ having $b$ bits of precision, we can compress PrivUnit/PrivUnitG to small seed length with a relatively efficient algorithm for seed expansion.

We will rely on Nisan's generator [33] which says that any space $S$ computation that consumes $N$ bits of randomness can be $\delta$-fooled using a random seed of length $O(\log N(S + \log N/\delta))$. Moreover, the computational cost of generating a pseudorandom string from a random seed is $O(N \log N)$. In our set up, the test that privacy of PrivUnit/PrivUnitG depends on is the $[g \cdot x \geq \gamma]$, when $g$ is chosen from the Gaussian distribution. This probability that this test passes for the Gaussian distribution is $e^{-c\varepsilon}$ for some constant $c$, and thus it suffices to set $\delta$ to be $e^{-c\varepsilon}\beta$ to ensure that mechanism satisfies $(\varepsilon + 2\beta)$-DP. For the rest of this discussion, we will set $\beta = \varepsilon\tau/2$ which leads to $\varepsilon' < \varepsilon(1 + \tau)$. We can set $\tau$ to be inverse polynomial as the dependence of the parameters on $\tau$ will be logarithmic.

The test of interest for us can be implemented in $S = O(\log d + b)$ space, and requires $N = db$ bits of randomness. Plugging in these values and $\delta = \varepsilon\tau e^{-c\varepsilon}/2$, we get seed length $O(\log db(b + c\varepsilon + \log db/\varepsilon\tau)$ and each expansion from seed to value requires run time $O(db \log db)$. Each run of PrivUnitG requires $O(e^{c\varepsilon})$ expected random strings, leading to a run time of $O(e^{c\varepsilon}bd \log db)$.

For our algorithm, we run this on a $k$-dimensional vector instead of a $d$-dimensional one, with $b = \log d$. This gives us seed length $O(\log(k \log d) \cdot (\log d + \varepsilon + \log(k \log d/\varepsilon)))$. Given the projected vector, the run time is $O(e^{c\varepsilon}k \log^2 d)$.