# OpenReview forum: "Fast Optimal Locally Private Mean Estimation via Random Projections"
_NeurIPS.cc/2023/Conference — NeurIPS 2023 poster_

### Official Review · Reviewer_rS8z · 2023-07-01

**Soundness:** 4 excellent
**Presentation:** 4 excellent
**Contribution:** 3 good
**Rating:** 6
**Confidence:** 3

**Summary:**

This paper presents an efficient distributed mean estimation framework, which first reduces local data dimension using (different) random projections and then applies an existing private processor (Asi et al [7]) on the projected data before sending them to the server. Further speedups of the framework are also presented, using a more efficient projection matrix and correlated sampling. Utility bounds of the framework and its speedups are derived, and experimental results on both synthesis and real-world data sets are presented.

**Strengths:**

[1] Accelerating private distributed mean estimation is a useful topic.

[2] The paper (especially, narrative) is beautifully written.

[3] Utility bounds are derived.

**Weaknesses:**

[1] Novelty of the proposed framework is a bit limited, as it basically applies random projection (in a non-traditional but still considerably straightforward way) on PrivUnitG (Asi et al [7]) to speed up the latter.

[2] Comparison of efficiency to the naive PrivUnitG in Table 2 is missing and should be added. Importantly, I suspect the proposed framework does not significantly improve over PrivUnitG, because it simply applies PrivUniteG on fewer features -- this is partly evidenced in the experimental results (Figure 3), where PrivUnitG appears more efficient.

[3] There is a potential error in the proof of Theorem 1 that may make the result invalid.

In line 486 (supplementary), equation (i) is argued to follow because $E[\hat{u}] = u$, but such argument does not make sense to me. Consider a simple case when n = 1, then the equation basically argues $E[||A + B||_2^2] = E[||A||_2^2] + E[||B||_2^2]$. I don't see how this can fly... Can authors elaborate?


---------- Update ----------

I've read the author responses and my concerns [2][3] are well-addressed. I have raised my score accordingly.

**Questions:**

Could authors respond to my comments [3] and [2] in the Weakness section?

---

> ### Author Rebuttal · Authors · 2023-08-09
>
> We thank the reviewer for their feedback. Below we answer all of their questions.
>
> 1. Novelty: While the techniques we use are similar to techniques used in the *non-private* version of the problem, our paper is the first to use them with privacy and several challenges arise in doing so. One of the technical challenges that comes from privacy is that the private randomizers only apply to vectors of bounded norm, thus requiring us to deterministically and carefully control the norm of the projected vector. While one could use a conservative upper bound on the norm (e.g. twice the expected norm),  the norm bound linearly impacts the privacy error, and this conflicts with our goal of getting (1+o(1))*OPT error.  We discuss these differences in lines 70-78 and we will elaborate and highlight them more prominently in the revised version.
>
>
> 2. Comparison to PrivUnitG: note that PrivUnitG has the same guarantees as PrivUnit in Table 2 (up to (1+o(1)) term in utility) and we will add a row for PrivUnitG to the table. Crucially, the main improvement of our algorithms over PrivUnitG is in communication complexity, where PrivUnitG requires a communication budget of d whereas our algorithms require epsilon*log(d), which is an exponential improvement. As we mention in the paper, our algorithm is the first communication-efficient algorithm for LDP mean estimation that obtains optimal utility bounds (with exact constants).
>
> 3. Error in proof of theorem 1: the proof does not contain an error. The reason is that $E[W_i \hat{u_i} - W_i u_i]=0$ since $E[\hat{u_i}] = u_i$ and this holds conditioned on $W_i$. Translating this to the reviewer’s example, this means that we have $E[A]=0$ conditioned on any $B=b$, and therefore $E[\| A+B\|^2] = E[\|A\|^2] + E[\|B\|^2]$. The  cross term vanishes since $E[\langle A,B\rangle] = E[\sum_i A_i B_i] = \sum_i E[A_i B_i] = \sum_i \sum_b \Pr(B=b) b \underbrace{E[A_i \mid B=b]}_0 = 0$. We will add further clarification to the proof in the paper. We hope this resolves the concerns that the reviewer has about the validity of our results.

---

> > ### Comment · Reviewer_rS8z · 2023-08-17
> > **Response by Reviewer**
> >
> > Dear authors,
> >
> > Thank you for clarifying the proof of Theorem 1 and efficiency comparison with PrivUnitG. My concerns on these two issues are well addressed. In line 486, it might be helpful to indicate what the expectation is with respect to, e.g., $W_i$'s and $\hat{u}_i$'s.
> >
> > Regarding novelty, I agree the theorectical analysis has challenges, but still think the proposed technique resembles its non-private counterpart to a great extent -- but perhaps it is common in DP-related research. Anyway, I've raised my score.

---

### Official Review · Reviewer_kcoe · 2023-07-05

**Soundness:** 3 good
**Presentation:** 3 good
**Contribution:** 3 good
**Rating:** 6
**Confidence:** 5

**Summary:**

The paper introduces a fast algorithm for private mean estimation under communication constraints. The proposed scheme primarily relies on random projection, and the algorithm enhances speed through the utilization of Hadamard matrix and sampling techniques.

**Strengths:**

- Theorem 1 offers a theoretical analysis of the utility achieved when random projection is employed.

- It is intriguing to observe that the combination of SRHT (Hadamard matrix + Rademacher diagonal matrix) achieves results close to the optimal.

- The idea of enhancing performance through correlated sampling appears to be a compelling approach.

**Weaknesses:**

1. I believe the current work primarily focuses on the tradeoff between utility and complexity, where the communication budget is relatively higher compared to other algorithms like SQKR. For instance, SQKR has a communication budget of epsilon (~10), while the proposed scheme has k on the order of 10^2 to 10^3 (the communication budget k\log^2 d is even larger).

2. In Figure 1, comparing the proposed scheme with a communication budget of k\log^2 d to SQKR with a communication budget of epsilon may appear unfair (despite being the optimal choices for their respective algorithms). To properly compare with SQKR, I suggest the authors provide two separate figures: one with a communication budget of k and another with a communication budget of k\log^2 d.

**Questions:**

- Could you please specify the exact value of k used in Figure 1?

- As both SQKR and the current work utilize Kashin's representation, which can be approximated using an orthogonal matrix, it might be worth exploring the possibility of a unified framework that encompasses both algorithms.

- Does the LDP constraint automatically hold due to the inclusion of PrivUnit in the projected space?
Even in correlated sampling?

**Limitations:**

Although the proposed work is impressive in terms of complexity, the communication budget is order-wise suboptimal compared to previous work (e.g., SQKR).

---

> ### Author Rebuttal · Authors · 2023-08-09
>
> We thank the reviewer for their positive feedback and comments. Below we answer all of their questions.
>
> 1. Comparison to SQRK (communication):
> The primary emphasis in this paper is on getting the optimal error bounds with practical communication:  our work is the first to achieve optimal utility with feasible communication that is logarithmic in the dimension. It's important to highlight that SQKR, regardless of the communication budget, fails to provide optimal utility bounds and exhibits an order of magnitude disparity in error (increasing k > epsilon doesn't reduce the error of SQKR). We note, however, that getting constant-optimal bounds with smaller communication is a natural open question.
> Additionally, SQKR relies on the model of shared randomness, which is less practical. In the private randomness model SQKR would additionally need to send k \log d bits to specify the indices of bits (as we show in Table 1).
>
> 2. Figure 1: we use k=1000 (see details in section 4 in the paper).
>
> 3. Kashin representation: we note that our work, unlike SQKR, does not use Kashin representation and instead builds on orthogonal transformations. Kashin representation has two main limitations compared to our approach: first, it is more computationally expensive to find a Kashin representation than using our transforms (e.g. SRHT), moreover, Kashin requires us to increase the dimensionality by a constant factor, hence, is unlikely to give utility bounds with optimal constants. Indeed, the difference comes from us requiring only an $\ell_2$ bound on the projected vector, whereas SQKR needs an $\ell_\infty$ bound, leading to additional constant overheads.
>
>
> 4. LDP constraint: the reviewer is right; privacy holds because we apply PrivUnit over a unit input vector. The transformation only changes the input vector, which doesn’t matter for the privacy analysis, hence this holds for input-independent correlated transformations as well (we will clarify this in the paper).

---

> > ### Comment · Reviewer_kcoe · 2023-08-21
> >
> > Thank you to the authors for their thorough response and elucidations. I have decided to increase my score.

---

### Official Review · Reviewer_r6Zt · 2023-07-11

**Soundness:** 4 excellent
**Presentation:** 3 good
**Contribution:** 3 good
**Rating:** 7
**Confidence:** 4

**Summary:**

The paper studies the problem of locally private mean estimation, and design algorithms with optimal (upto $1+o(1)$ factor) error with better communication complexity and runtime. The key idea behind improved communication complexity and runtime is (a certain) random projection based pre-processing step.

**Strengths:**

1. The problem of private mean estimation is an important problem in itself and moreover, a lot of interest is also derived from its application to federated learning. Hence, there has been a lot of work recently on devising "better" algorithms for it in various settings. The paper identifies gaps in our understanding of the problem, mainly on obtaining optimal error with better communication complexity and runtime, and make solid contributions towards them.

2. The paper is written well. Instead of presenting the final result first, it gradually builds towards conveying the challenges and key ideas.

3. The experiments are encouraging, testing various aspects and comparison with many known methods. Further, the experiments section is well-organized and outlined.

**Weaknesses:**

1. The underlying key technique -- random-projection to smaller dimensions -- is simple and has been used in the prior works for this problem (albeit without the privUnit). Further, even improvements gained via Subsampled Randomized Hadamard Transform follow standard ideas used in prior works. I encourage the authors to highlight if there are significant differences between this paper and the prior works.

2. While I enjoyed reading Section 2 where the authors gradually build towards the main results and various instantiations, I found Section 3 to be short. It does not provide intuition on how correlated sample is beneficial. Similarly, Remark 2 references unbiased versions of the proposed procedures, deferred to Appendix A.  I encourage the authors to expand on these, if possible.

Minor: Table 2 referenced is missing in the main text.

**Questions:**

Please respond to the weaknesses. Another simple questions is: why not use same $W$ in every client? Then the server won't have apply the inverse transformation (matrix multiplication) for each client. Is there some fundamental reason this isn't done?

**Limitations:**

The paper focuses on obtaining optimal (upto upto $1+o(1)$ factor) error for worst-case instances (minimax risk). Consequently, methods which achieve optimal error but with (large) constant factor are deemed inferior. However, in practice, the instances are usually not worst-case but are rather structured (eg: sparse) and therein non-optimal methods may fare better (unlesss there is something fundamental about the achieving worse error regardless of the instance). Therefore, more instance-specific guarantees may be needed to reflect improvement in practice. This is not to say that worst-case error (which is studied in this paper) isn't interesting and important.

---

> ### Author Rebuttal · Authors · 2023-08-09
>
> We thank the reviewer for their detailed and positive feedback. Below we answer all of their questions.
>
> 1. Technical challenges: While the techniques we use are similar to techniques used in the *non-private* version of the problem, our paper is the first to use them with privacy and several challenges arise in doing so. One of the technical challenges that comes from privacy is that the private randomizers only apply to vectors of bounded norm, thus requiring us to deterministically and carefully control the norm of the projected vector. While one could use a conservative upper bound on the norm (e.g. twice the expected norm),  the norm bound linearly impacts the privacy error, and this conflicts with our goal of getting (1+o(1))*OPT error.  We discuss these differences in lines 70-78 and we will elaborate and highlight them more prominently in the revised version.
>
> 2. Unclarity in Section 3: as requested by the reviewer, we will add more intuitions and clarifications to section 3. Briefly, the main point of that section is to improve the server runtime compared to non-correlated transformations: the protocols of section 2 (non-correlated transformations) have runtime ~ n*d whereas the protocols of section 3 (based on correlated transformation) have a significantly better runtime of ~ n + d.
>
> 3. Same matrix for each client: unfortunately, using the same matrix for all clients is not possible in our setting as our goal is to estimate the mean, hence using the same matrix for all clients results in large bias for mean estimation. Section 3 uses correlated matrices, which allows to improve the runtime of the server, requiring only a single inverse step in this case. We will elaborate on this more in the paper.
>
> 4. Worst-case vs instance-dependent: we agree with the reviewer about the importance of obtaining instance-dependent guarantees. However, the focus of our paper is on the worst-case setting, and we leave the instance-dependent setting to future work.

---

> > ### Comment · Reviewer_r6Zt · 2023-08-21
> > **Thanks!**
> >
> > I thank the authors for their detailed response and clarifications. I am still positive about the paper, and will keep my score.

---

### Official Review · Reviewer_Dc2U · 2023-07-12

**Soundness:** 4 excellent
**Presentation:** 4 excellent
**Contribution:** 3 good
**Rating:** 7
**Confidence:** 4

**Summary:**

The paper considers the problem of performing mean estimation for worst-case data under local differential privacy. The precise setting considered involves $n$ users, each of whom has one data vector $v_i$ (which, without loss of generality, can be assumed to exist within the unit ball). The users must privatize their data and send them to a central server which tries to estimate $\frac{1}{n} \sum_i v_i$ as accurately as possible using the privatized data. The older work of Asi et. al. showed that the PrivUnit algorithm of Bhowmick et. al. attains the optimal error rate among all unbiased estimators. However, implementing that approach leads to concerns regarding the number of bits that need to be exchanged, as well as the runtime, both from the perspective of the clients and the server. The present work tries to address these concerns by using an approach relying on random projections. In particular, the users embed their data into a low-dimensional sub-space of the original space and privatize the resulting vectors. This results in a significantly smaller number of bits being exchanged, thus addressing the issue of potential constraints in communication. In greater detail, the authors give two different algorithms, both of which fit into the above framework. The first algorithm involves each user randomly sampling their own projection matrix, which they must communicate to the server. On the other hand, the second algorithm involves the server choosing a matrix which they communicate to the clients, and then each client has to transform the matrix using their own sampling matrix. For both approaches, the overhead in the error is small ($1 + o(1)$ specifically), while the latter approach manages to significantly reduce not just the communication between servers, but also the runtime for both the clients and the server. The theoretical results are complemented by experiments which support them.

**Strengths:**

The paper is very well written and gives a complete picture, covering both theoretical aspects and empirical evaluation. Additionally, the coverage of the prior work in the introduction is quite thorough, so the results are clearly positioned within the overall literature. Considering the computational aspects (in addition to the statistical ones) of performing mean estimation under local differential privacy is a very well motivated question, which the paper does a good job of addressing by giving algorithms that have significant improvements compared to prior work. For that reason, I believe that it is of interest to the NeurIPS community.

**Weaknesses:**

No obvious weaknesses.

**Questions:**

No questions.

**Limitations:**

No obvious negative societal impact.

---

> ### Author Rebuttal · Authors · 2023-08-09
>
> We thank the reviewer for their positive feedback.

---

### Decision · Program_Chairs · 2023-09-21

**Decision:**

Accept (poster)

**Comment:**

This paper addresses the problem of distributed mean estimation with privacy. The recent optimal algorithms of PrivUnit suffers from large communication cost. This paper significantly improves the number of bits required to be transmitted, using random projections. The reviewers agree that this paper makes foundational advances in a fundamental problem of private mean estimation.